# A spatio-temporal analysis investigating completeness and inequalities of global urban building data in OpenStreetMap

Benjamin Herfort [1,2] ✉, Sven Lautenbach [1], João Porto de Albuquerque [3], Jennings Anderson [4] & Alexander Zipf [1,2]

OpenStreetMap (OSM) has evolved as a popular dataset for global urban analyses, such as assessing progress towards the Sustainable Development Goals. However, many analyses do not account for the uneven spatial coverage of existing data. We employ a machine-learning model to infer the completeness of OSM building stock data for 13,189 urban agglomerations worldwide. For 1,848 urban centres (16% of the urban population), OSM building footprint data exceeds 80% completeness, but completeness remains lower than 20% for 9,163 cities (48% of the urban population). Although OSM data inequalities have recently receded, partially as a result of humanitarian mapping efforts, a complex unequal pattern of spatial biases remains, which vary across various human development index groups, population sizes and geographic regions. Based on these results, we provide recommendations for data producers and urban analysts to manage the uneven coverage of OSM data, as well as a framework to support the assessment of completeness biases.

Between 2001 and 2018 the urban population growth and built-up area expansion accelerated, especially in large cities in the low- and lower-middle-income countries as defined by World Bank[1]. Global urban land expansion is expected to grow rapidly in the next 20 years[2]. Buildings constitute one of the most important physical elements of urban settlements[3]. However, little is known on a consistent basis about building inventories worldwide; a spatially detailed survey of the distribution and concentration of the building stock does not yet exist[4]. Improving the systematic monitoring of the global urbanization process is a requirement for achieving the United Nation's Sustainable Development Goals (SDGs), e.g. urban SDG 11 (Make cities and human settlements inclusive, safe, resilient and sustainable.), whilst especially in low-income countries the data are usually scarce[1].

Although building data are usually maintained by national statistics offices, they are underfunded by an estimated gap of $1 billion USD globally and consequently, baseline geospatial data that should be provided by these agencies are often not accessible, not up-to-date or not available in standard formats[5,6]. Tackling building data scarcity requires moving beyond insufficient traditional data sources to utilizing non-traditional sources for measuring the SDGs[7,8].

Building data is an essential asset in global urban analyses for assessing progress towards a number of important urban goals. For instance, SDG Indicator 11.3.1 (ratio of land consumption rate to population growth rate) would directly benefit from building footprint data. However this indicator is currently mainly based on easily available remote sensing data, e.g. World Settlement Footprint (WSF)[4], which exacerbates the monitoring of structural changes such as changes in floorspace per capita or re-densification trends. The monitoring of SDG Indicator 11.1.1 (proportion of urban population living in slums, informal settlements or inadequate housing) would benefit from an analysis of building blocks and street networks considering their spatial relations, such as density and neighbourhood relations[9]. SDG Indicator 11.7.1 (average share of the built-up area of cities that is open space for public use for all, by sex, age and persons with disabilities) can be partly derived from Sentinel-2 satellite data, but this data source falls short for distinguishing public from private green

[1]Heidelberg Institute for Geoinformation Technology, Heidelberg, Germany. [2]GIScience Chair, Institute of Geography, Heidelberg University, Heidelberg, Germany. [3]Urban Big Data Centre, University of Glasgow, Glasgow, United Kingdom. [4]Meta Platforms Inc., Salem, OR, USA. ✉e-mail: benjamin.herfort@heigit.org

spaces as it fails to capture fine-grained urban structures due to its 10 metres resolution[10]. Global Building Morphology Indicators could allow us to quantify the form of urban areas and enable analyses comparing morphological parameters across cities which go much further than existing approaches which are often based on aggregated population data[11]. In addition to these examples of urban research using global building footprint data, Table 1 provides recent policy examples and practical applications of global urban analysis using building footprint data.

It has been shown that open data communities - such as OpenStreetMap (OSM) - are already contributing to filling existing data gaps[7,12,13]. OSM is now used widely for applications such as web maps and navigation services and OSM data has been used in domains such as urban planning[14], SDG monitoring[15], disaster management[13,16], public health[17–19], as well as for supporting crisis response during the

COVID-19 pandemic[20]. In addition to contributions by individual volunteers (mappers), there is an intensifying trend of organized corporate and humanitarian mapping communities contributing to OSM[12,21]. So far, corporations have predominantly focused on mapping roads in OSM, but they are also interested in building data, as testified by current efforts towards automated identification based on machine learning, such as the Microsoft Building open dataset[22]. As these datasets have been generated outside OSM but released with an OSM-compatible format, the two data sources could be feasibly combined[23]. In contrast, data on roads have been predominantly directly edited within OSM by corporations, as they require significantly more effort to conflate, needing to maintain referential integrity and a topological network to operate properly.

However, particular attention needs to be paid to data inequalities and biases when OSM data is utilized in global urban studies or to

**Table 1 | Policy and practice examples and applications of global urban analysis using building footprint data**

| Domain | Stakeholder | Description |
|---|---|---|
| Urban Sustainable Development (SDG 11) | UN Habitat | Building footprints and street network layout enable the detection of informal settlements via topological analysis, which can be used to inform infrastructure extensions with a focus on low and middle-income countries[9]. UN Habitat is using this for global monitoring of the SDG Indicator 11.1.1: Proportion of urban population living in slums, informal settlements or inadequate housing[76]. |
| | UN Habitat | UN Habitat is using global building footprints to assess the spatial structure, size and population distribution of city blocks to improve walkability and access to the public transport system. This is related to global monitoring of the SDG Indicator 11.2.1: Proportion of population that has convenient access to public transport, by sex, age and persons with disabilities[77]. |
| | UN Habitat | Building footprints in combination with other features such as roads and specific land use types can be used to model land consumption rates. This is used by UN Habitat for global monitoring of the SDG Indicator 11.3.1: Ratio of land consumption rate to population growth rate[78]. |
| Public Health (SDG 3) | WorldPop | Building footprints enable a precise disaggregation of population datasets at granular spatial scale to estimate where people live at the household level. This is done in the WorldPop programme and has had numerous application cases in support of public health interventions[2]. |
| | Clinton Health Access Initiative | In the Clinton Health Access Initiative Malaria elimination campaign, using OSM data field teams were able to get a full overview of the buildings they needed to cover for Malaria interventions such as spraying campaigns[79]. |
| | Médecins Sans Frontiéres (MSF) | During the 2014 Ebola response in Guinea MSF used building footprints, roads and other infrastructure from OSM to identify and support affected communities for the first time on a large scale. As of 2023 OSM is the reference geographical dataset for most of MSF operations on the ground[80]. |
| Disaster Risk Reduction and Climate Adaptation (SDG 13 and Sendai Framework) | World Bank Open Data for Resilience Initiative (OpenDRI) | Building footprints are needed to estimate exposure and vulnerability to climate-related hazards and disasters when conducting infrastructure and network risk assessment at large scales, including national and globally. OpenDRI created building footprint data in OSM and engaged local government, civil society, and the private sector in disaster risk reduction strategies with a focus in subsaharan Africa, Asia and more recently Latin America and Caribbean[81]. |
| | RiskFactor | FloodFactor uses building footprints (from OSM and other sources) to help project past, present, and future flood risk at the household level across the United States. Building footprints are utilized in a similar way to predict fire risk (FireFactor) and heat risk (HeatFactor)[82]. |
| | Pacific Disaster Center | The DisasterAWARE platform uses many data layers from OSM (including building footprints) to support nongovernmental and governmental organizations worldwide when planning disaster risk reduction strategies and developing integrated early warning and decision support system[83]. |
| Humanitarian Action and Emergency Response | Copernicus Emergency Mapping Service (EMS) | Building footprints and roads are used in Copernicus EMS rapid disaster response maps and data products to provide information about affected population and infrastructure. This supports first responders in their specific crisis management tasks[84]. |
| | UN Department of Operational Support, UN Global Service Center | UN Mappers is using OSM building footprints to enrich topographic and operational data in support of UN peacekeeping and humanitarian missions worldwide[85]. |
| | Humanitarian OpenStreetMap Team (HOT), UN OCHA | HOT mobilizes online volunteers around the world to map buildings, roads and other features in OSM in response to disasters. Daily updated extracts from OSM are provided to first responders through UN OCHA's Humanitarian Data Exchange (HDX) platform[86]. |

derive global data products to inform policy and decision-making. Since the OSM data is generated by volunteers who are not evenly distributed around the globe, their mapping efforts in OSM have been strongly biased towards high-income countries (HICS)[24,25], even if the humanitarian mapping community has contributed to spread efforts towards some cities in low- and middle-income countries (LMIC) to some extent[12]. We use the World Bank's country classification, where LMICs have incomes of $1,086-$13,205 per capita and HICS above the threshold.

A common data quality requirement for many research and policy applications based on building footprint data is the completeness of the building stock for analysis purposes. This is particularly important for comparative analyses that seek to discern global urban patterns, such as for instance to derive a global dataset of critical infrastructure[26], or to use big data for comparing urban morphology across the globe[27]. When unaccounted for, spatial bias in completeness can lead urban analysts and researchers to draw general conclusions which are only valid for well-represented (well-mapped) areas[28]. Completeness of building stock data is particularly important for ensuring equitable and fair decision-making based on OSM data for the policy and practice applications of Table 1 and, as such, has a direct importance for the overarching principle of the SDGs of leaving no one behind.

Previous studies assessed OSM compared to authoritative data to provide detailed insights on the completeness of OSM in selected cities[3,29]. However, more work is needed to transfer methods to other regions for which reference datasets are either missing or unavailable. To overcome dependencies on sparsely available administrative datasets, proxy data that is globally available such as remote sensing data (e.g. Nighttime Light, built-up-area, Sentinel 2 derived spectral indices)[30] or population data[31] have been suggested as a potential resource to assess and predict OSM building completeness.

Considering the widespread usage of OSM building footprint data for urban analyses and policy-making, here we investigate building data completeness and inequality in OSM on the global scale for for 13,189 urban centres around the world, which are home to an estimated population of 3.5 billion people (about 50% of the global population). Our spatio-temporal analysis pursues the following two research questions:

1. What is the completeness of OpenStreetMap building data in the context of global urban analysis applications?
2. How unequal is urban OpenStreetMap building data distributed within the space of a city, across continents and on the global level?

First, we propose a machine-learning regression method based on a random forest to assess OSM building completeness within 13,189 urban centers (as defined by the European Commission[32]). We utilize an extensive collection of open building data from commercial and authoritative sources as training data and utilize OSM full-history data for spatio-temporal data analysis on the global scale[33]. The model further relies on information obtained from remote sensing data (land cover, population distribution, nighttime lights), Subnational Human Development Index (SHDI), and urban road network density as predictors. Second, this paper builds upon the extensive methodological skill set developed to investigate urban segregation and transfers it towards analysing geographic inequalities within OSM. This allowed us to present a comprehensive assessment of the evolution of urban OSM building completeness, which encompasses all data contributed to OSM since 2008.

## Results
### Urban OSM building completeness
Our results reveal that for 1,848 cities (14% of the analysed amount) OSM building footprint data exceeded 80% completeness. In total,

**Table 2 | OSM building completeness in urban centers on the global scale and grouped by world regions, Subnational Human Development Index class and city size class measured by population**

| | n | Completeness [%] | Humanitarian Mapping [%] | Corporate Mapping [%] |
|---|---|---|---|---|
| **Global** | **13,189** | **24** | **10.0** | **0.1** |
| **World Regions** | | | | |
| East Asia & Pacific | 3068 | 20 | 13.4 | 0.2 |
| Europe & Central Asia | 1351 | 71 | 1.1 | 0.1 |
| Latin America & Caribbean | 1073 | 20 | 15.2 | 0.4 |
| Middle East & North Africa | 901 | 12 | 16.9 | 0.2 |
| North America | 378 | 64 | 0.2 | <0.1 |
| South Asia | 3997 | 9 | 17.8 | <0.1 |
| Sub-Saharan Africa | 2421 | 30 | 51.1 | 0.1 |
| **Subnational Human Development Index** | | | | |
| Low | 2289 | 28 | 52.3 | 0.1 |
| Medium | 4960 | 15 | 30.3 | 0.3 |
| High | 3883 | 17 | 15.8 | 0.2 |
| Very High | 1967 | 59 | 2.1 | <0.1 |
| **City Size by Population** | | | | |
| Small Urban Areas | 10,930 | 23 | 10.1 | 0.1 |
| Medium-Size Urban Areas | 1348 | 32 | 9.5 | <0.1 |
| Metropolitan Areas | 563 | 37 | 9.4 | 0.1 |
| Large Metropolitan Areas | 287 | 41 | 10.4 | 0.1 |

Completeness was computed as the average of the individual OSM building completeness values per urban center. Humanitarian mapping and corporate mapping were quantified by their share on the overall map data. SHDI classes were based on cut-off points defined by the United Nations Development Programme[68]: low human development (SHDI< 0.550), medium human development (SHDI: 0.550 - 0.699), high human development (SHDI: 0.700–0.799), very high human development (SHDI> 0.800). City size classes were based on population thresholds defined by OECD[69]: small urban areas (50k-200k), medium-size urban areas (200k–500k), metropolitan areas (500k–1.5M), large metropolitan areas (>1.5M).

these cities were home to a population of 492 Million people (16% of the global urban population). Contrary, our results show that for 9,163 cities (69% of the analysed amount and home to 48% of the global urban population) OSM building footprint data did not reach 20% completeness. Our analysis found a global average urban OSM building completeness of 24% per urban center (see Table 2). Relatively high completeness was estimated for Europe & Central Asia (71%) as well as for North America (64%). Completeness values lower than the global average were observed for the regions Latin America & Caribbean (20%), East Asia & Pacific (20%), Middle East & North Africa (12%), and South Asia (9%). The completeness value for East Asia & Pacific was strongly influenced by the fact that urban centers in China were hardly mapped, as legislation prevents OSM mapping. Sub-Saharan Africa completeness (30%) was slightly higher than the global mean.

We found that organized humanitarian mapping activities in urban centers contributed an average of about 10% of the building footprints globally (see Table 2). However, humanitarian contributions were focused on specific regions, especially in Africa where more than 50% of all building edits in Sub-Saharan Africa were related to organized humanitarian mapping activities. Overall, organized

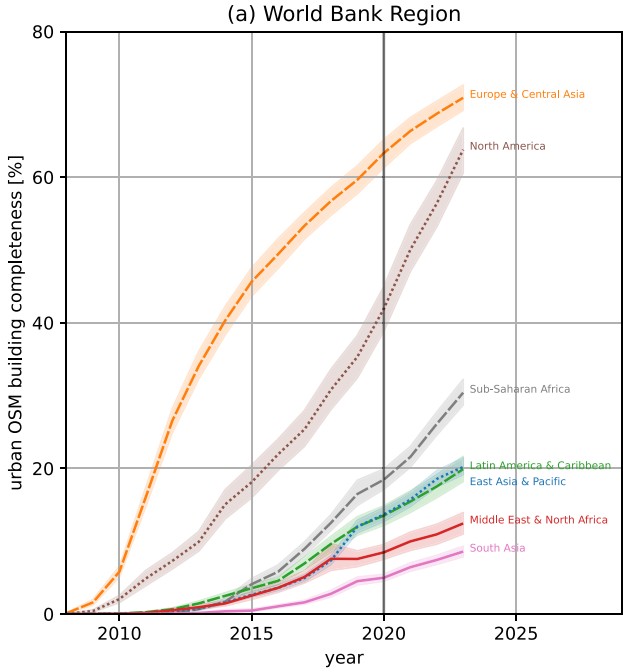

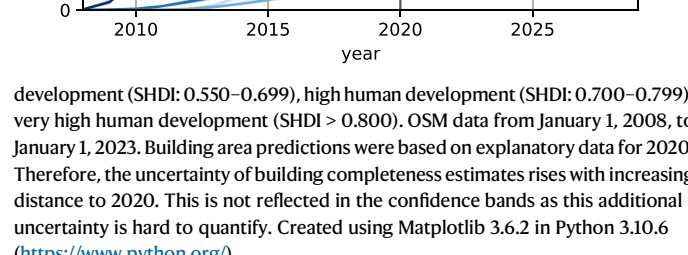

**Fig. 1 | Temporal evolution of urban OSM building completeness.** Average values are derived for **a** world regions and **b** Subnational Human Development Index (SHDI) group. Completeness was derived by aggregating building area predictions based on a Random Forests model and annual OSM building area per urban center. The shaded areas represent the 95% confidence interval for each line. SHDI classes were based on cut-off points defined by the United Nations Development Programme[68]: low human development (SHDI < 0.550), medium human development (SHDI: 0.550–0.699), high human development (SHDI: 0.700–0.799), very high human development (SHDI > 0.800). OSM data from January 1, 2008, to January 1, 2023. Building area predictions were based on explanatory data for 2020. Therefore, the uncertainty of building completeness estimates rises with increasing distance to 2020. This is not reflected in the confidence bands as this additional uncertainty is hard to quantify. Created using Matplotlib 3.6.2 in Python 3.10.6 (https://www.python.org/).

humanitarian mapping activities were expectedly associated with lower subnational human development index values, in line with previous findings[12]. We generally found corporate mapping activity to constitute less than 2% of all building edits globally (and only about 0.1% in urban centers), a significant difference in participation from corporate mappers editing nearly 20% of the global road network as previously found[21].

Distinguishing urban centers by SHDI also revealed dramatic differences in the temporal trajectories of completeness (see Fig. 1 (b)). In general, urban centers in regions with very high SHDI had the highest levels of mapped building completeness. Surprisingly, however, there was no positive correlation between SHDI and completeness. The completeness in low SHDI urban centers was higher than the completeness of urban centers with high SHDI. Our results suggest that this was due to the positive impact of organized humanitarian mapping activities since 2015, especially on urban centers located in low and medium SHDI regions (see Table 2).

The size of the urban centers measured by population was positively correlated to completeness (see Table 2), albeit the differences were not as pronounced as for world regions or SHDI classes. OSM building data in large metropolitan areas were considerably more complete compared to small urban areas. However, the temporal evolution of urban building completeness showed very similar patterns for urban centers regardless of their population (see Supplementary Figure 1).

The spatial distribution of building completeness across urban centers shows a strong regional variability across that global trend: numerous cities in any region were mapped with a very high completeness regardless of the overall completeness or mapping activity in that region (see Fig. 2). For instance, within Africa, we found urban centers in Egypt and Ethiopia with particularly low OSM building completeness, whereas cities in Tanzania, Uganda and western African countries achieved much higher completeness. Similarly, building

completeness values in Indonesia and the Philippines were notably higher than for other countries of Southeast Asia. In contrast, most urban centers in India and China were hardly mapped with regard to building footprints. Strikingly, the spatial distribution of OSM building completeness for urban centers was characterized by spatially clustered patterns at various scales.

The uneven building completeness between urban centers was also indicated by a global Gini coefficient of 0.8. This characteristic was observed across all regions and was most pronounced in South Asia and Sub-Saharan Africa (c.f. Fig. 3 a). The temporal evolution of the Gini coefficient indicated that both globally and regionally, OSM building data distribution has become slightly more even over time. In contrast, we find that global spatial inequality in OSM building completeness sharply increased between 2008 and 2014 (c.f. Fig. 3 b). During that time-although overall OSM building completeness became more even (as measured by the Gini coefficient)-mappers favoured cities near already well-mapped cities. We also find that until 2014, the expansion of OSM mapping to distant and un-mapped regions (likely to be located in the LMIC) did not happen at a significant scale.

Nevertheless, since 2014, Moran's I as a measure of global spatial autocorrelation declined from 0.71 till 0.56 as of 2023. This indicated that the spatially clustered completeness pattern became less intense, albeit still clearly visible. Combined with the decrease of the Gini coefficient in the same period, our results suggests OSM building data in 2023 was much less segregated in both evenness and clustering compared to the state-of-the-map in 2014. Attention should be paid to the fact that Gini coefficient and Moran's *I* have been stagnating on the global level since 2019, which might indicate a shift in mapping behaviour due to restrictions caused by the COVID-19 pandemic. However, since 2021 Gini coefficient started to increase again, implying a tendency towards a more uneven distribution of building mapping across urban centers on the global level.

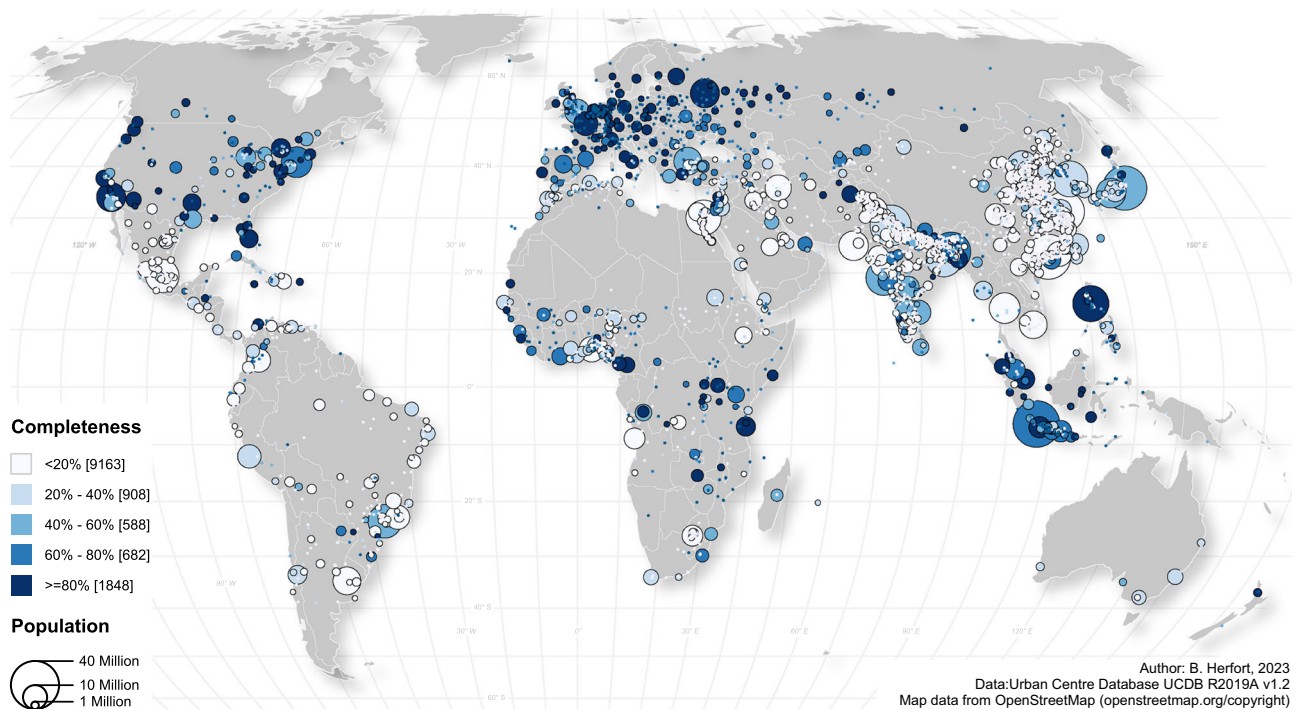

**Fig. 2 | Spatial distribution of OSM building completeness in 13,189 urban centers.** For each class the overall number of urban centers is reported in the squared bracket. For an interactive web map visualization visit https://hex.ohsome.org/#/urban_building_completeness. OSM data as of 2023-01-01. Created using QGIS 3.28.3 (https://www.qgis.org/en/site/).

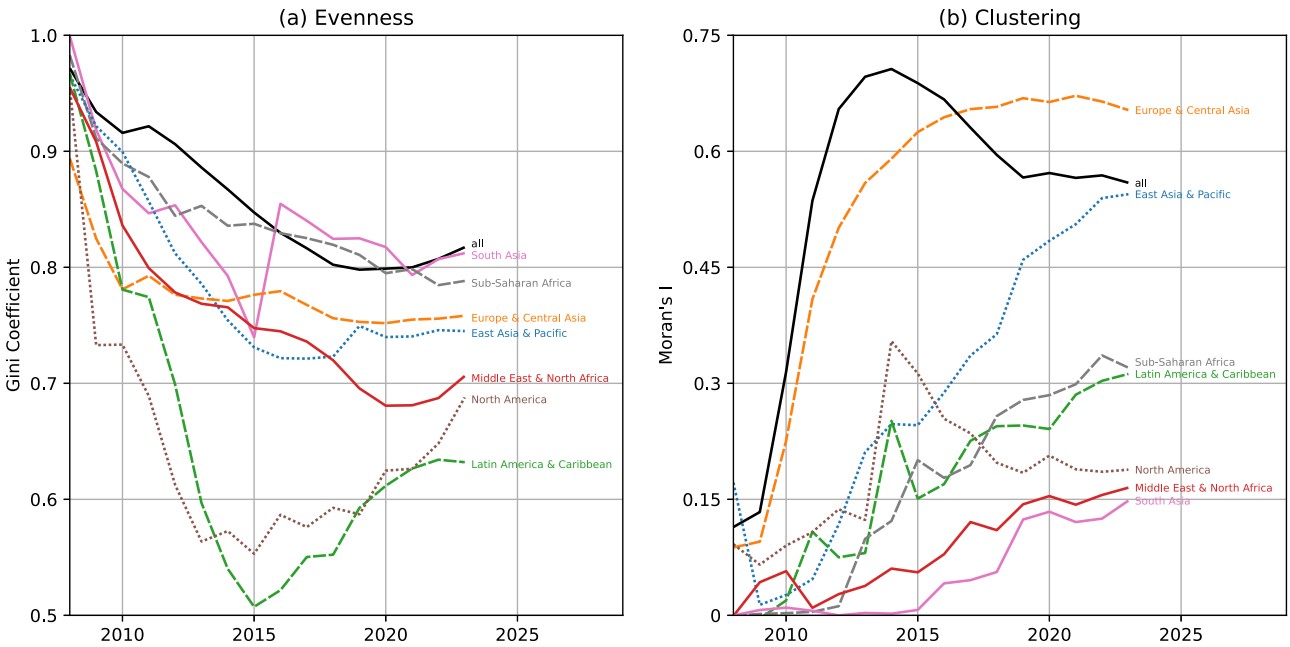

**Fig. 3 | Nonspatial and spatial inequality measures of completeness.** Temporal evolution of (**a**) evenness and (**b**) clustering of urban OSM building completeness per world region. Moran's *I* measures spatial autocorrelation, positive values indicate spatial clustering. Values for Moran's *I* in practice often range between -0.5 and 1.15 with zero indicating absence of global spatial autocorrelation. OSM data from 2008-01-01 to 2023-01-01. Created using Matplotlib 3.6.2 in Python 3.10.6 (https://www.python.org/).

In support of the global pattern, we also found spatial autocorrelation within regions to increase steadily over time regardless of overall map completeness. Europe & Central Asia reveal a moderate spatial clustering (Moran's *I*: 0.23) in 2010, but a very high spatial clustering (Moran's *I*: 0.65) in 2023 (see Fig. 4a and b). In 2023, urban centers with high completeness in Europe & Central Asia were surrounded by other urban centers with high completeness and this effect was much stronger than in 2014. An analogous process was observed

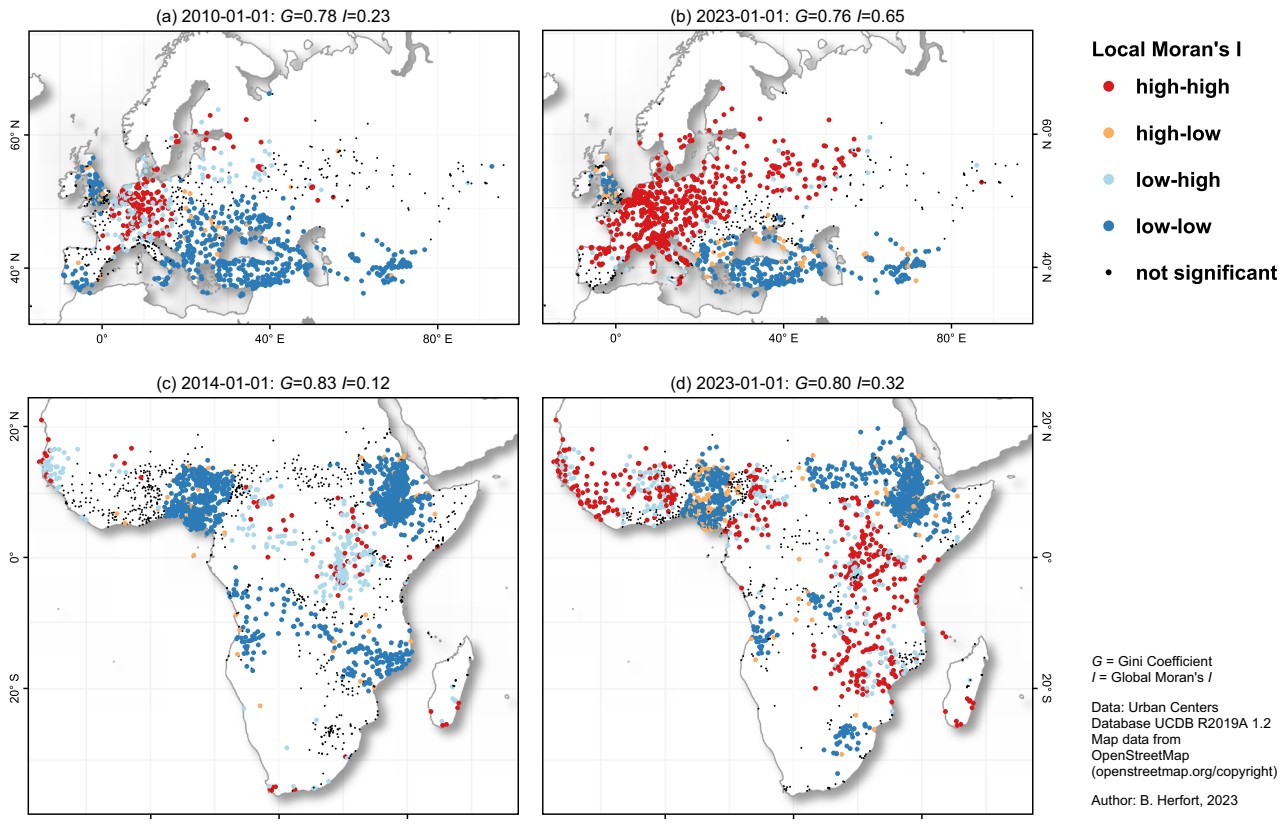

**Fig. 4 | Local spatial autocorrelation of completeness.** A comparison at two points in time for urban centers within **a**, **b** Europe & Central Asia and **c**, **d** Sub-Saharan Africa. Each urban center was classified according to whether its building completeness value was above (high) or below (low) the global mean and if the weighted mean across its neighbors was above or below the global mean. Based on this, four quadrants are defined: high-high (HH), low-high (LH), low-low (LL) and high-low (HL). High-high describes clusters of high completeness values, low-low describes clusters of low completeness values while low-high and high-low indicate spatial outliers in the sense that the completeness value of the urban area was unexpected in their neighborhood. Significance levels were adjusted for multiple testing. For each region and point in time we provide the Gini coefficient (G) and Moran's I for the region shown in the sub-plot. Created using QGIS 3.28.3 (https://www.qgis.org/en/site/).

in all other regions, e.g. as shown for Sub-Saharan Africa in Fig. 4c and d. The only exception to this is North America, where spatial clustering of building completeness has decreased since 2014.

## Intra-Urban OSM Building Completeness

This section builds upon the intra-urban spatial heterogeneity of OSM building completeness estimated at a resolution of one square kilometer. Here we will refer to each one square kilometer section of the map as a grid cell. These grid cells were used to calculate the Gini coefficient and Moran's I for individual urban centers. Based on both those indicators and OSM building completeness, urban centers were classified into three different types (and dividing into two additional sub-types) utilizing an agglomerative clustering approach (c.f. Fig. 5 and c.f. Fig. 6). Figure 7 shows the temporal evolution or OSM building completeness per urban center and highlights the trajectory for each cluster representative. The spatial distribution of urban centers based on the intra-urban inequality (see Supplementary Figure 2) shows a similar spatial pattern as Fig. 2.

Urban centers of type (1) (see cluster dendrogram in Fig. 6) showed very low completeness and could be further distinguished into subtypes (1a) and (1b), respectively. Urban centers of type (1a) could appear as white spots on the map, more aptly described as the unmapped cities and are characterized by a high Gini coefficient and very low Moran's I value. There was no particular spatial pattern defining where the small number of eventually mapped buildings would be located within the city. Among 1,692 urban centers, the city of Faisalabad, Pakistan is shown in Fig. 6 (1a) as an example of this

category. Urban centers of type (1b) could be considered hardly mapped as well, but there were often a few grid cells which have been mapped with a much higher completeness in regard to buildings. Such mapped grid cells were not distributed randomly, but tended to cluster spatially as indicated by a higher Moran's I value. The urban agglomeration of Guadalajara, Mexico (see Fig. 6 (1b)) exemplifies that there were several distinct mapping hot spots surrounded by a larger number of unmapped grid cells.

The second group of urban centers is characterized by relatively high Moran's I and moderate to high evenness covering cities with wide range of completeness. For urban centers of type (2), the overall completeness value hardly reflected the local completeness values. The urban centers of type (2a) such as Las Vegas, USA (see Fig. 6) could be considered a divided city from the perspective of mapped building in OSM. This pattern is characterized by a few spatially clustered neighbourhoods which are mapped very well, whereas large parts of the city remain unmapped. Also for urban centers of type (2b) a highly segregated spatial distribution of OSM building completeness was observed. For these cities, such as Abidjan, Ivory Coast, there exist large blocks of completely mapped grid cells, however still entire neighbourhoods of the city are missing from the map.

Finally, urban centers with the highest overall completeness and very low spatial clustering were most likely to get classified as type (3). As shown in Paris, France (see Fig. 6 (3a)), almost all parts of the city may be considered completely mapped. Only a few grid cells remained unmapped and these are often not strongly spatially clustered. Figure 7 highlights that for the case of Paris between 2010 and 2013

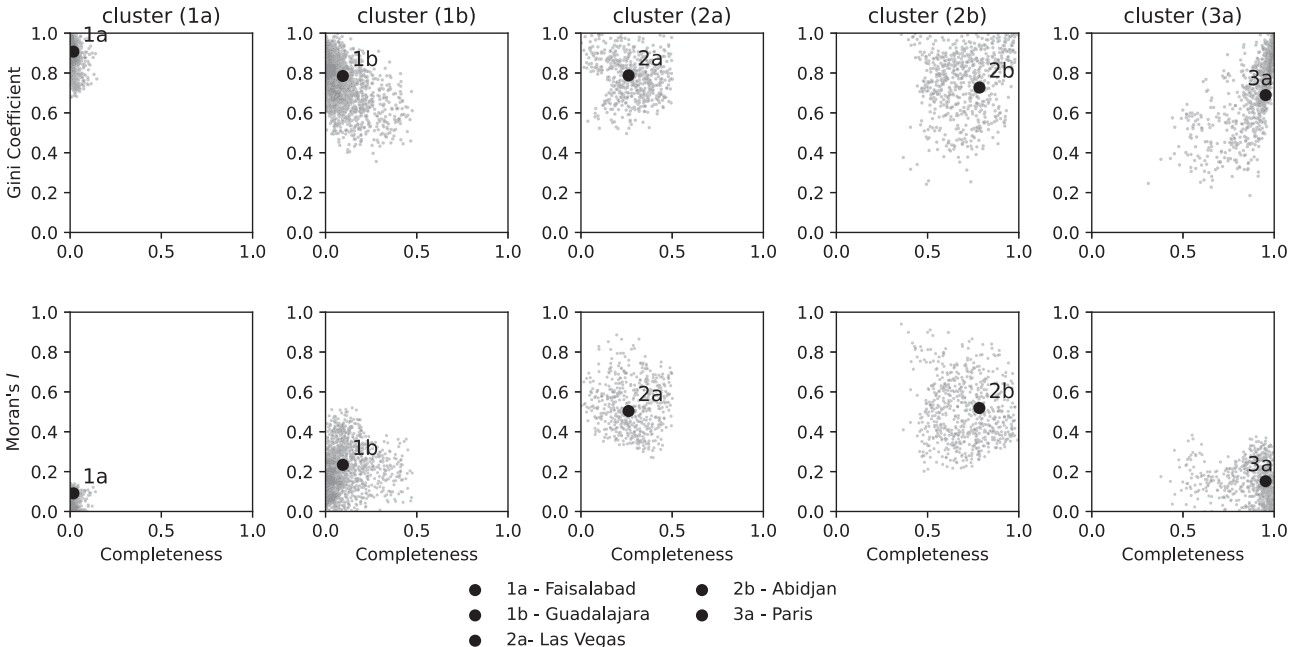

**Fig. 5 | Agglomerative clustering of urban centers based on OSM building completeness, Gini coefficient *G* and Moran's *I*.** Each point represents a single urban center with a minimum area of 25 square kilometers. Smaller urban centers were ignored as Gini coefficient and Moran's *I* could not be reliably estimated. For each of the clusters a single representative example was selected out of the 4,647 urban centers considered in this analysis. OSM data as of 2023-01-01. Created using Matplotlib 3.6.2 in Python 3.10.6 (https://www.python.org/).

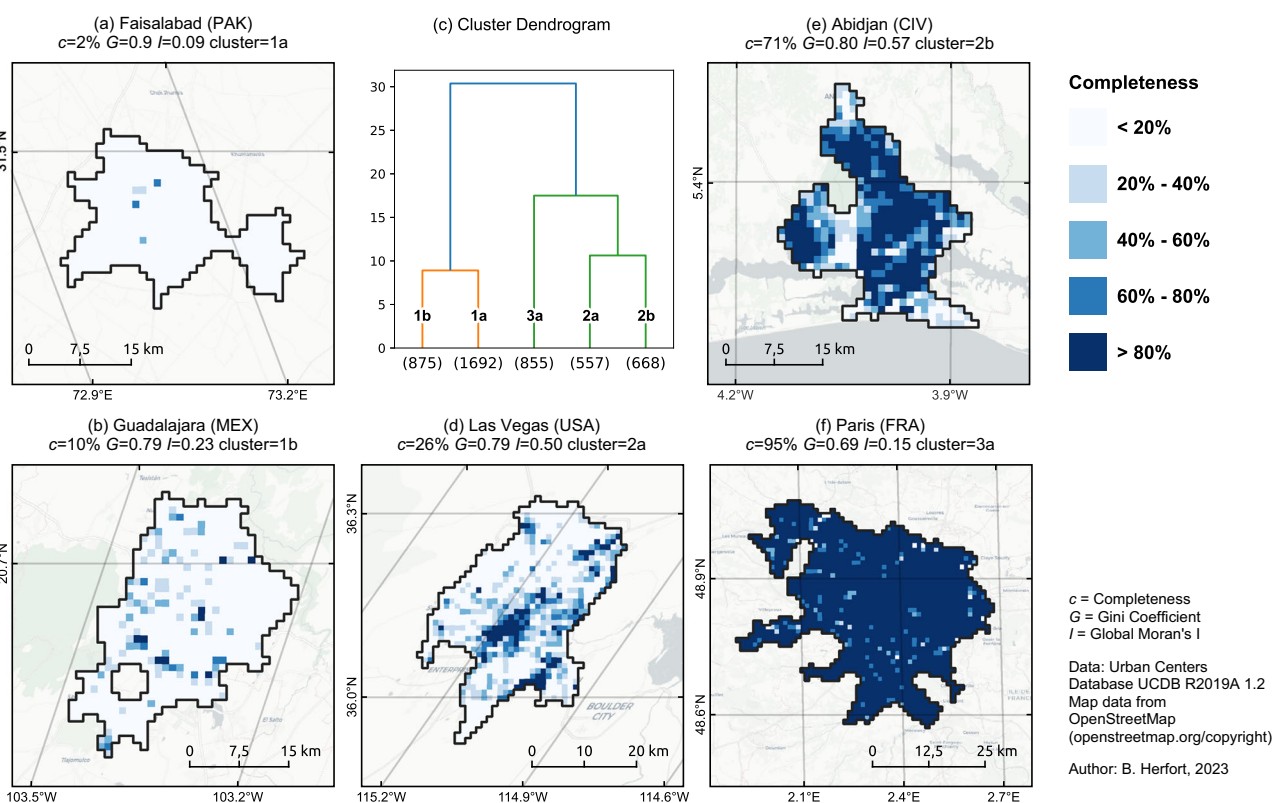

**Fig. 6 | Intra-urban OSM building completeness.** Spatial distribution for selected urban centers (**a**–**f**). For each urban center we report on overall OSM completeness *c*, Gini coefficient *G* and Moran's I. Cell size is always one square kilometer for any urban center. The clusters are the same as in Fig. 5. The number of urban centers in each cluster is indicated in the dendrogram (**b**). For an interactive web map visualization visit https://hex.ohsome.org/#/urban_building_completeness. OSM data as of January 1, 2023. Created using QGIS 3.28.3 (https://www.qgis.org/en/site/) and Matplotlib 3.6.2 in Python 3.10.6 (https://www.python.org/).

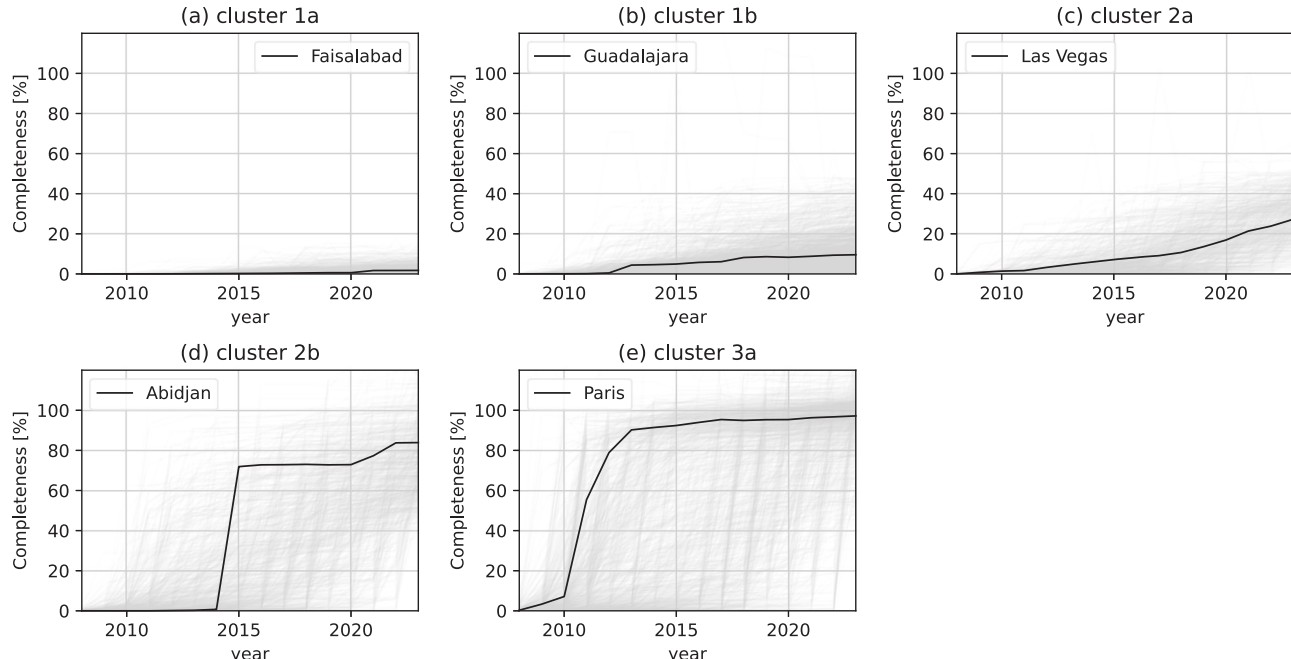

**Fig. 7 | Urban center level temporal evolution of OSM building completeness.** We report on completeness per cluster and for selected urban centers (**a**–**e**). The clusters are the same as in Fig. 5. OSM data from January 1, 2008, to January 1, 2023. Created using Matplotlib 3.6.2 in Python 3.10.6 (https://www.python.org/).

almost 80% of the entire building stock was added to OSM. As opposed to type (1), the urban centers of type (3) could be described as the well-mapped cities and represent some finality of mapping building footprints in OSM.

## Discussion

Mapping efforts of communities in OSM over the previous decade have made OSM a unique global database of building footprint data, which is accessible to all with no licensing costs. Besides the well-examined bias in OSM between high-income countries and low- and middle-income countries, our results demonstrate that the OSM building stock as of 2023 showed a much more spatially diverse pattern than previously considered, which was shaped by regional, socio-economic and demographic factors across several scales. The results confirm the similar, albeit less dichotomous, global completeness pattern of land use land cover (LULC) information from OSM[34] and stand in slight contrast to the relatively higher and more evenly distributed completeness for OSM's road network[35]. This highlights the complexity of OSM mapping activities and the challenge for urban analyses based on OSM data. Thus, we want to provide two important recommendations for OSM data users and producers to pick the best strategies to cope with OSM's uneven spatial coverage and to promote the sustained impact of future mapping efforts.

First, the unequal patterns of completeness we evidenced make it essential for urban analysts to assess the potential negative impact of missing data, i.e., OSM data users should investigate if the intended urban analysis is subject to spatial bias caused by OSM's uneven spatial coverage at multiple scales. To support this assessment, we provide an open dataset[36] resulting from this study with estimated completeness maps for 13,189 urban agglomerations worldwide using a grid which enables the assessment of variations within and across urban centres.

We encourage researchers and practitioners to further develop this datasets and the other data quality assessment methods, which may help to quantify the highly uneven geographies of participation in communities such as OSM or as observed in Wikipedia[37]. Global studies and global frameworks will benefit from these approaches, as researchers will be able to draw more robust conclusions and avoid misleading recommendations for decision-makers once the biases in

OSM's coverage are known and can be accounted for. This will further increase the reliability and usefulness of using OSM data for global urban analysis such as monitoring progress towards the SDG's[15], as well as for humanitarian activities, including disaster risk reduction[13].

Second, the global community of OSM data producers will be able to use our completeness maps as a guidance to plan where future mapping activities should take place to improve coverage so that no one is left behind as encouraged by the SDGs. By combining the completeness maps with socio-demographic characteristics of the areas of interest, it will be possible to ensure a fair and balanced selection of target geographic areas to reduce existing data inequalities within OSM on the global, regional or intra-urban level. Our open dataset about urban OSM building completeness can inform this process. In light of the critical challenges to finance high-quality data systems for addressing inequalities in SDG monitoring in both low and middle-income countries[38]-despite heightened demand[39]-the creation and usage of OSM data could be promoted further as a cost-effective alternative. Data generation with OSM not only allows filling the current data gaps essential to monitor progress, but can also be an pathway for equitable urban transformations by empowering local communities to have a voice and benefit from the data production process[40]. The cultural openness and social nature of OSM is a clear strength to achieve transparency about existing inequalities and how to address them, especially in comparison with building footprint datasets that are derived using proprietary, black-box machine learning approaches for which bias and fairness are often still unknown[41].

However, our analysis also comes with unavoidable limitations that need to be considered. One major limitation is that the work presented here only investigated buildings mapped in OSM within urban centers. Even if our study encompasses about 50% of the global population which lives in urban agglomerations, one should be careful to transfer our findings to rural areas. Researchers have shown that there is a tendency of OSM data to be of higher quality in cities[3]. On the other hand, rural areas that have been the target of humanitarian mapping campaigns will likely have higher completeness than surrounding areas[42].

The results of machine learning models, such as the random forest model used by us to derive building area predictions, are conditioned

**Table 3 | Global and regional model performance measures based upon 20-fold spatial cross-validation**

| region | n | $r^2$ | exp var | MSE | MAE |
|---|---|---|---|---|---|
| **Building Area Prediction (1km Grid)** | | | | | |
| Global | 403,357 | 0.74 | 0.74 | 0.0025 | 0.034 |
| East Asia & Pacific | 48,389 | 0.72 | 0.73 | 0.0028 | 0.038 |
| Europe & Central Asia | 76,714 | 0.73 | 0.73 | 0.0019 | 0.029 |
| Latin America & Caribbean | 51,792 | 0.70 | 0.70 | 0.0047 | 0.047 |
| Middle East & North Africa | 31,748 | 0.77 | 0.77 | 0.0027 | 0.037 |
| North America | 99,064 | 0.70 | 0.70 | 0.0012 | 0.025 |
| South Asia | 55,743 | 0.84 | 0.84 | 0.0022 | 0.030 |
| Sub-Saharan Africa | 39,907 | 0.67 | 0.67 | 0.0024 | 0.035 |
| **OSM Building Completeness Prediction (Urban Centers)** | | | | | |
| Global | 6553 | 0.9 | 0.9 | 0.012 | 0.055 |
| East Asia & Pacific | 483 | 0.93 | 0.93 | 0.009 | 0.050 |
| Europe & Central Asia | 1103 | 0.85 | 0.85 | 0.016 | 0.089 |
| Latin America & Caribbean | 874 | 0.88 | 0.88 | 0.011 | 0.044 |
| Middle East & North Africa | 663 | 0.90 | 0.90 | 0.005 | 0.024 |
| North America | 367 | 0.92 | 0.92 | 0.009 | 0.067 |
| South Asia | 1660 | 0.88 | 0.88 | 0.007 | 0.024 |
| Sub-Saharan Africa | 1403 | 0.81 | 0.83 | 0.034 | 0.091 |

by potential biases present in the training data or biases that arise from the algorithms[41]. For the authoritative data utilized to train the model, a high geometric accuracy was assumed, however, these datasets might be outdated depending on the publisher's update cycle. The quality of building footprint data from Microsoft (see Supplementary Table 1) showed a sufficient recall of more than 80% for urban centers in all regions. However, self-declared recall values by Microsoft have been considerably lower for some regions. This indicates that recall might be lower in rural areas, which have not been included into our study. Nevertheless, the remaining biases of the Microsoft buildings dataset for urban areas are also reflected in the results reported here and might lead to extra low building area predictions and consequently over-estimated OSM building completeness. We were also not able to quantify the uncertainty for countries with a large number of urban centers (e.g. China) for which training data was not available. Nevertheless, other authors have highlighted that rapidly urbanizing cities, for instance in China[43], are mimicking suburbanisation trends and patterns of the post-World-War-II United States. As such, we do not expect severe structural deviations in respect to the modelled relationships between explanatory variables and building area prediction for these countries. The low feature importance for the geographic world region (i.e., variable World Bank Region Code) seems to be in line with that assumption.

To be transparent about model uncertainty, we reported on the model performance utilizing a spatial cross-validation procedure (c.f. Table 3). While the completeness estimation performed well with a global $r^2$ score of 0.9, slightly higher uncertainty was observed for Sub-Saharan Africa. However, this may constitute a starting point for local communities and researchers to design local completeness models that can overcome the limits of the global modelling approach used in this study, even if these local models might not be easily transferred to other regions.

Our findings extend the general pattern of urban OSM building completeness as of 2020-01[44] a) by highlighting the temporal evolution of mapping activity in conjunction with other events such as the COVID-19 pandemic, b) by considering a spatially much more balanced and extensive training data set from various sources and c) by investigating the global pattern and consequences of inequalities in completeness.

In addition, geospatial data quality is comprised of dimensions beyond measuring completeness[45]. For some sectors, such as public health programs, assessment of completeness is only the first step, and information on building usage is also required, but often only available for a small subset of buildings[18]. Recent work has outlined pathways towards regional and global scale analysis of the quality of building attribute data from OSM and other sources[46,47]. It has been shown for land cover and land use (LULC) in OSM that the spatial pattern for completeness and accuracy are not necessarily the same[34]. Whereas urban OSM building completeness and OSM LULC completeness show similar global trends on the national level, OSM building data accuracy needs to be further investigated, e.g. towards its potential utilization as training and/or validation samples in machine learning models. A more detailed analysis of the temporal trajectories for urban centers would facilitate this investigation and might reveal to what extent data quality dimensions, temporal evolution of mapping activity and inequality measures are mutually dependent.

Future work should further investigate the potential of a harmonious ensemble dataset that combines the best of OpenStreetMap buildings with additional building coverage from deep learning-based datasets such from Microsoft Buildings[22] or from official sources[47]. We have performed the direct comparison between OSM and Microsoft buildings and report the findings in the Supplementary Materials section (see Supplementary Table 2 and Supplementary Figure 3). Ultimately, OSM buildings and MS buildings represent two very different types of data, and the major advantage of OSM is that it is continuously verified by human editors. This is the reason why not all of the Microsoft buildings are in OSM, as the Microsoft buildings represent a different type of dataset: algorithmically extracted from aerial imagery using automated methods. For some places, the building footprints in OSM might already come from an imported dataset, but they need to be accepted by a human-in-the-loop process, for instance using the mapwith.ai editor[48] or similar tools. A large direct import of Microsoft building data into OSM seems unlikely at the moment, however, it has been shown for the road network that imports can increase contributor activity especially for already engaged mappers and an interplay of data imports and updates by contributors could improve OSM data significantly[49,50].

The OSM community started its journey in the early 2000's in Western European cities and this history is still clearly visible in the unbalanced spatial distribution of map data in 2023. Nevertheless, there are numerous successful examples of local mapping communities that are overcoming structural barriers which exclude others from participating in OSM. We believe that by empowering these communities, OSM will further evolve into the most comprehensive open geographic data base which is needed to help achieve the SDGs and ensure equitable and sustainable urban futures.

## Methods
### Building data
The analysis was carried out for 13,189 urban centers on a global scale. We used the Global Human Settlement Layer Urban Centres Database (GHS-UCBD) developed by the European Commission to delineate our study areas[32]. Accordingly, urban centers have been characterized as high-density clusters of contiguous grid cells of 1 km$^2$ with a density of at least 1500 inhabitants per km$^2$ and a minimum population of 50,000[32]. Each urban center was spatially disaggregated using a one square kilometer grid based on the equal-area Mollweide projection. The grid adopted the same structure utilized by the raster datasets of the GHS-UCBD. The grid cells are not always squared, as they deviate from a perfect rectangular 1x1 km shape depending on latitude and longitude of each grid cell. The shape distortion adds uncertainty to our results for a very small number of urban centers. These are located in very low and very high latitudes which are also far away from the Greenwich meridian, e.g. in New Zealand. For each of the resulting

665,641 grid cells, we aggregated both the reference data sets (if available) and the datasets utilized as predictors in the model.

We derived the overall OSM building footprint area in square kilometers for each grid cell using the ohsome API, which relies on the OSHDB framework for spatio-temporal analysis of OSM history data[33]. We included buildings which have been mapped in OSM as of 2023-01-01. We considered all polygon OSM objects tagged with building=*.

As no single reference building data set on the global scale exists, we combined a set of external datasets (c.f. Supplementary Table 3) which have been obtained either from authoritative or commercial sources building upon the great work by Biljecki et al. (2021)[51]. We derived the overall building footprint area in square kilometers for each grid cell in the reference datasets by intersecting the grid cells and building footprints and then summing up the corresponding surface area of all building (parts) per grid cell. In cases where two reference datasets were available, e.g. from both Microsoft and an authoritative source, we considered only the information from the authoritative dataset. In total, these reference datasets covered 6,633 urban centers (404,982 grid cells) across 162 countries. In some regions, data was only available for selecrted cirties, not the entire country. Microsoft's data was derived using a deep learning-based building detection approach[22].

The Geo-Wiki built-up reference dataset[52] has been utilized to assess the suitability of the Microsoft building footprint dataset for our OSM completeness modelling approach. The Geo-Wiki campaign visually assessed very high-resolution satellite images of 50K sample locations for the presence of built-up surfaces containing any building with a roof using a crowdsourcing approach[52]. All Geo-Wiki grid cells intersecting the urban center geometries for which Microsoft building data was available at the city level were considered in the analysis. Precision and recall for the Microsoft building footprints have been derived on the 10x10 meters Geo-Wiki grid level (see Supplementary Table 1). Grid cells were defined as true positives (TP) if at least one Microsoft building footprint intersected with a Geo-Wiki grid cell labelled as Built-up. True negatives (TN) were defined as Geo-Wiki grid cells labelled as Not built-up that did not intersect with any Microsoft building footprint. Accordingly we defined false positives (FP) as grid cell labelled as Not built-up which intersected with at least one Microsoft building and false negatives (FN) as the reverse. Finally, we also provide precision and recall for the Microsoft building footprints as self-declared by Microsoft in the Global ML Building Footprints GitHub repository (see Supplementary Table 4).

### Explanatory variables

As explanatory variables we used the following datasets (see Supplementary Table 5): the Global Human Settlement Layer Population (GHS-POP) is provided by the European Commission[53] and based on a disaggregation of CIESIN's Gridded Population of the World (GPWv4.10). We utilized the Subnational Human Development Database to characterize regions based on their socio-economic status[54]. We relied on the aggregated SHDI and did not further consider its individual components (education, standard of living, health) in our analysis. Information on night-time lights was obtained as the annual average of 2020 and aggregated by summing up all values per grid cell[55]. Land cover information at 10 meter resolution was utilized from the ESA WorldCover 2020 dataset which has been derived from Sentinel-1 and Sentinel-2 data[56]. We derived the overall area per land cover class for each grid cell in square kilometers.

We extracted the road network length in kilometers from OSM per grid cell for main roads. Main roads were selected using the ohsome API[33] with the following filter: highway in (primary, primary_link, secondary, secondary_link, tertiary, tertiary_link, unclassified, residential). We included all data mapped in OSM as of 2023-01-01. We have investigated the spatial variations in the completeness of OSM road data by utilizing an intrinsic quality assessment approach following the

mapping saturation methodology proposed by Rehrl & Gröchenig (2016)[57]. Accordingly, the completeness of the road network was estimated for each urban center and aggregated by world region (see Supplementary Table 6). The intrinsic completeness measure revealed that for a small, but still decent share of the urban centers in South Asia (10.5%), Middle East & North Africa (10.2%) and Sub-Saharan Africa (5.9%) road network mapping could be considered only in the initial stage. It is very likely that the majority of the road network is not (yet) completely mapped for these urban centers.

The data values for ESA WorldCover, GHS-POP and VNL are subject to uncertainty, as they reflect the situation for the year 2020, albeit they are utilized in a model to assess OSM building completeness for a time range from 2008 to 2023. The model will be most suited to predict the building stock of 2020 and not the past building stock, nor the correct building stock for 2023. As a consequence, the analysis will slightly overestimate completeness for regions which have seen rapid urbanisation since 2020, but might also underestimate completeness for timestamps before 2020.

Initially, we considered additional explanatory variables (e.g. permanent water bodies, fossil fuel consumption, OSM railway length, OSM amenity count), but these have been disregarded as their feature importance in the Random Forest model turned out to be very low (<0.02).

### Building area prediction model

We used a Random Forest (RF) regressor[58] to predict the overall building area in square meters per grid cell using the covariates described in the section above. Building completeness is not directly predicted, but inferred in a second step using this prediction and the corresponding surface area of all OSM buildings per grid cell. In general, there are various applications of Random Forest regressors for producing spatial predictions[59], as well as estimating building completeness in particular (e.g. in Haiti, Dominica and St. Lucia[30]). RF constitutes a non-spatial approach to spatial prediction as sampling locations are ignored during the calculation of the model parameters[59]. Hence, in this study, we initially also considered generalized additive models (GAMs) as an explicit spatially aware approach (if smooths of coordinates are included) which has been used for geospatial modelling e.g. in the domains of geomorphology[60] or for the analysis of social media data[61]. Nevertheless, our results revealed that RF outperformed the GAM approach and we decided to utilize the RF implementation in the Python package scikit-learn[62].

To evaluate the performance of the proposed building area prediction we adopted a spatial cross-validation approach based on k-means clustering. Especially for large-scale mapping studies, such as this work, but also in the domain of ecological modelling data are almost always spatially autocorrelated and a spatially explicit assessment of machine learning models is required[63]. Due to spatial autocorrelation in the observations (data from nearby locations will not be independent), training samples and validation samples cannot be randomly selected as this would lead to overly optimistic error estimates[64]. Spatial blocking of samples, e.g. through k-means clustering, decreases this spatial dependence and provides more realistic performance scores[64,65]. Our spatial cross-validation blocks were derived using a 20-fold k-means clustering based on scikit-learn's python implementation[62].

We investigated the performance of our model in respect to building area prediction for the grid and in respect to OSM building completeness prediction for the urban centers using the following indicators: $r^2$ score, explained variance, mean squared error (MSE) and mean absolute error (MAE). On the grid level, MAE describes the average of the absolute differences between predicted building area and reference building area obtained from authoritative sources or Microsoft's Global ML Building footprints in square kilometers. Accordingly, MSE describes the average of the squared differences

between the predicted building area and reference building area per grid cell. On the urban centers level, MAE describes the average of the absolute differences between predicted building completeness and reference building completeness. MSE also describes the average of the squared differences between predicted building completeness and reference building completeness per urban center. These measures were calculated based on scikit-learn's python implementation[62].

To report global performance, we first estimated the model performance metrics for each of the seven regions (see Supplementary Table 2). As each of these regions contained different numbers of samples, we computed the global scores from the weighted average of the regional scores using the total number of samples per region as the weight. Table 3 provides the performance scores. Overall, the OSM building completeness model performed with a mean absolute error of 0.067 and achieved a $r^2$ score of 0.84 and explained variance of 0.85. In addition, we checked for spatial clustering in the residuals of the OSM building completeness prediction utilizing Moran's $I$ as a measure of spatial autocorrelation[66]. The residuals were not distributed entirely random across space, but nevertheless only showed a slight tendency to cluster (Moran's $I$: 0.29).

The distribution of raw residuals resembled a normal distribution for all regions (see Supplementary Figure 4). The histogram of raw residuals revealed that for samples located in Sub-Saharan Africa and North America, the distribution was slightly negatively skewed and had a weak tendency to predict too low completeness values for urban centers. For some regions, e.g. East Asia & Pacific or Sub-Saharan Africa, we observed heteroscedasticity in the distribution of residuals for urban centers, meaning that the higher the predicted completeness value, the larger the residual and thus the uncertainty of our model. Still, our predictions are conservative in the sense that they are underestimating the completeness for urban centers.

## Urban OSM building completeness

We calculated the OSM building completeness for each urban center using the area ratio method (reference building area / OSM building area), which has been applied by several other researchers in the context of urban areas[3,67]. We did not consider the building count, but building area instead, to account for the high sensitivity to disparities in modelling when using unit-based completeness measures[3], which is especially important when using a wide range of different building datasets as in this study. For instance, some datasets, such as OS OpenMap Local, model terraced housing as a single polygon, whereas in OSM these buildings are usually subdivided into multiple features. While this would result in a slight difference in the surface area of all buildings per grid cell, the building count can vary dramatically and completeness might be overestimated.

First, we obtained the overall predicted building area by summing up the values for all grid cells per urban center. OSM building completeness per urban center was derived annually by computing the ratio of OSM building area versus predicted building area. We report on the average monthly OSM building completeness for urban centers globally and distinguished this score further by world region, SHDI class and city size class by population. For the spatial aggregation we used the regions defined by World Bank and will refer to them also as world regions in the manuscript. In addition, 95% confidence intervals have been calculated for each time series. SHDI classes were based on cut-off points defined by the United Nations Development Programme[68]: low human development (SHDI< 0.550), medium human development (SHDI: 0.550–0.699), high human development (SHDI: 0.700–0.799), very high human development (SHDI> 0.800). City size classes were based on population thresholds defined by OECD[69]: small urban areas (50k–200k), medium-size urban areas (200k–500k), metropolitan areas (500k–1.5M), large metropolitan areas (>1.5M).

We investigated the impact of humanitarian mapping through the HOT Tasking Manager and corporate mapping by Apple, Meta,

MapBox, Microsoft and Kaart on overall completeness and inequality measures. OSM contributions have been considered as humanitarian mapping activities following the approach developed by Herfort et al. (2021), which utilizes information obtained from a HOT Tasking Manager database dump[12]. Corporate mapping activities were identified by OSM user ID, expanding on the approach presented in[21] by using a mapper's self-disclosed corporate affiliation in their OSM user bio instead of relying on potentially out-of-date lists on the OSM wiki[70]. According to these two approaches, the contributions of each OSM user were first categorized as either humanitarian, corporate or other, and in a second step according to world region, SDHI class and city size class by summing up the building area added to OSM per map edit. Based on this information, we derived the share of humanitarian map edits and corporate map edits on the overall OSM building data considering the building area (not building count) as the base unit.

Several measures have been adopted to describe the temporal evolution of inequality in urban OSM building mapping on the global scale and per world region. This analysis has been conducted for annual snapshots from 2008-01-01 up until 2023-01-01. The Gini coefficient has been utilized to derive the degree of evenness of urban OSM building completeness following an approach proposed by Massey & Denton (1988) to study residential segregation[71]. Analogous to their approach, the Gini coefficient was derived from the Lorenz curve, which plots the cumulative proportion of observed OSM building area against the cumulative proportion of missing building area (difference between OSM building area and predicted building area) across urban centers, which are ordered from smallest to largest proportion of observed building area. The Gini coefficient constitutes a non-spatial measure of segregation which provides insights on the evenness dimension, but does not allow conclusion about the spatial structure. Pysal's segregation package has been utilized to calculate the annual Gini coefficient from 2008-01-01 up until 2022-01-01[72].

Moran's $I$[66] has been selected as a measure of global spatial autocorrelation of urban OSM building completeness. A high Moran's $I$ value describes situations where urban centers and their neighbors showed similar high (or low) values of completeness. It's values are not strictly bound by the interval [-1,1]. The range depends on the largest and the smallest eigenvalue of the spatial weight matrix used, but frequently ranges from -0.5 to 1.15[73]. A Moran's $I$ value close to zero indicates a spatially random pattern, where the completeness of an urban center was not correlated to the completeness of its neighbours. Spatial autocorrelation has been proposed as an explicitly spatial indicator of segregation covering the dimension of clustering[71,74]. Following this approach, a high degree of clustering describes a spatial structure where areas with OSM building mapping are contiguous and closely packed, creating a single large block of mapped urban centers. In contrast, a low level of clustering implies that the observed OSM building stock is widely scattered around the globe (or within regions)[71]. Moran's $I$ relies on the definition of the spatial weight matrix - here, it was defined based upon the centroid of each urban center using a distance band threshold of 5 degree and an inverse distance weighting with a power of 1. Weights were row standardized. The neighborhood definition led on average to 367 neighbours per urban center and 13 urban centers were classified as islands for which no neighbours were identified.

We used the same spatial weight matrix to calculate local spatial autocorrelation (Local Moran Statistics[75]) for Europe & Central Asia and Sub-Saharan Africa to compare spatial inequalities within these regions between two timestamps. Pysal's esda package was utilized to calculate the global and local Moran's $I$ statistics from 2008-01-01 up until 2022-01-01[72].

## Intraurban OSM building completeness

To ensure a sufficient sample size, we calculated both inequality measures (Gini coefficient, Moran's $I$) for the intra-urban assessment

only for urban centers with a minimum area of 25 square kilometers respective 25 data points. For the resulting 4,722 urban centers local OSM building completeness was derived using the area ratio method described above for each square kilometer grid cell. Similarly, the Gini coefficient and Moran's $I$ global spatial autocorrelation of the local OSM completeness were calculated per urban center as described above. The spatial weight matrix was defined based upon the Queen contiguity graph. As such grid cells that share at least a vertex were considered as neighbours. The weight matrix was row-standardized.

The investigation was complemented by an agglomerative hierarchical cluster analysis of urban centers considering evenness and clustering within each city. The number of clusters was selected based on the hierarchical structure of the full dendrogram (see Fig. 6) and by investigating the Variance Ratio Criterion (Calinski-Harabasz Index) and Silhouette Coefficient for various number of clusters (see Supplementary Table 7). This analysis revealed that a number of two or three clusters would be optimal when only considering the clustering performance scores. We decided for three main clusters (1-3), but also report subclusters for 1 (a, b) and 2 (a, b) to allow for a more fine-grained distinction of urban centers, especially for those with low or medium completeness.

The distance matrix was based on the euclidean distance between OSM completeness, Gini coefficient and Moran's $I$. Since all variables considered in this analysis already showed a similar range of values between 0-1 normalization was not necessary. The analysis was conducted based on scikit-learn's python implementation[62] using the ward linkage criterion. Results were displayed for each cluster using scatter plots.

Cluster representatives were selected by first calculating the average values for Moran's $I$, Gini coefficient and completeness across all urban centers per cluster. Based on these cluster centroids, the euclidean distance to each sample was derived. Among the 15 samples closest to the cluster centroid, one representative was selected per cluster.

### Reporting summary
Further information on research design is available in the Nature Portfolio Reporting Summary linked to this article.

## Data availability
Source data are provided with this paper. The full set of data generated in this study (e.g. for training and running the machine learning model and the final results presented in all figures and maps) have been deposited in the Figshare database under accession code https://doi.org/10.6084/m9.figshare.22217038. The Global Human Settlement Layer Urban Centres data used in this study is available from https://ghsl.jrc.ec.europa.eu/ghs_stat_ucdb2015mt_r2019a.php. Population information has been obtained from GHS-POP, which is available at https://jeodpp.jrc.ec.europa.eu/ftp/jrc-opendata/GHSL/GHS_POP_GLOBE_R2023A/GHS_POP_E2020_GLOBE_R2023A_54009_1000/V1-0/GHS_POP_E2020_GLOBE_R2023A_54009_1000_V1_0.zip. Subnational Human Development Index data can be retrieved from https://globaldatalab.org/shdi/download/. The ESA WorldCover dataset is available at https://zenodo.org/record/5571936. Information on night-time lights was obtained from https://eogdata.mines.edu/nighttime_light/annual/v20/2020/VNL_v2_npp_2020_global_vcmslcfg_c202101211500.average.tif.gz. The raw full-history planet OSM data can be downloaded from https://planet.openstreetmap.org/planet/full-history/.

## Code availability
All Python code and Jupyter notebooks necessary to calculate the geospatial statistics, create maps and derive figures are available in this GitHub repository: https://doi.org/10.6084/m9.figshare.22241776.

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

## Acknowledgements

The authors would like to thank the OSM contributors for their inspiring work, as well as Matthias Schaub, Levi Szamek, Rafael Troilo, Michael Auer and Clemens Langer for their great help with building the computational framework for the analysis. B.H. and S.L. were supported by the Klaus Tschira Stiftung.

## Author contributions

B.H., J.P.A. and S.L. conceived the analysis. B.H., S.L. and J.A. conducted the experiments and analyzed the results. B.H. and S.L. prepared graphics and tables. B.H. interpreted the data and wrote the paper with contributions from S.L., J.P.A., J.A. and A.Z.

## Funding

## Competing interests

S.L., J.A. and A.Z. declare no competing interests. B.H. and J.P.A. are unpaid voting members of the Humanitarian OpenStreetMap Team. Voting Members are responsible for voting "on matters affecting the Corporation including, but not limited to, the election of directors and [additional] voting members."
