## [Peer Review File · Nature Communications]

A spatio-temporal analysis investigating completeness and inequalities of global urban building data in OpenStreetMapEditorial Note: Parts of this Peer Review File have been redacted as indicated to remove third-party material where no permission to publish could be obtained.

REVIEWER COMMENTS

Reviewer #1 (Remarks to the Author):

The paper assesses OSM building completeness in a large number of urban agglomerations around the world. This is much needed work and the first such study doing so. The method is also novel, as typical OSM building completeness studies conducted at the national or city level cannot be scaled globally. The authors should be complimented on this important effort and valuable results. While to my understanding spatial data quality and OSM have not been subject of the journal, NC may be a suitable venue for it, and on the broader scope, it may also contribute to raising awareness of OSM among readers who may not be much into it.

While I am positive about this work and would like to see it published in NC, I have several suggestions for improvement, some major, while some minor. If I misunderstood some concepts, the authors are advised to make some of the explanations in the paper clearer and more prominent.

- It may be questionable what constitutes a global study. And this may not be a truly global study. This is not necessarily a flaw, as I understand that the authors have focused only on urban areas (which is also obvious in the title), and that makes sense. It's definitely a significant advancement since previous attempts, but "only" 50% of the global population is taken into account, while much of the narrative may suggest otherwise. The focus is on areas that contain 183m buildings, while OSM at the moment has about 3x more buildings mapped than that. The authors argue multiple times that nobody should be left behind. But aren't they doing so too? Why not include all areas? Does the method really not work on rural areas, i.e. everywhere? If not, then the narrative should be adjusted a bit. I am not sure that this story is sensible as they investigate the global divide, but at the same time they perpetuate it by compounding the divide between urban and rural areas.

- The paper does not discuss much on the actual use of such data, especially at the large or even global scale (besides using it for city-scale analyses in a single city), which may be needed to assert the importance of the work. They do focus on SDG, but that is just one of many applications of building footprints. There have been recent studies that use a large volume of building footprints beyond one city, in fact, some of them do so using OSM building data, such as urban form, energy, and climate studies, e.g. the global building morphology indicators project. A few more sentences about why such data is important and how can it be used wouldn't hurt.

- Somewhat in relation to the previous point, I have a comment pertaining to the first RQ 'Is OpenStreetMap building data good enough for urban analysis and SDG monitoring?' -- I am not sure the paper actually answers this RQ, at least adequately. There are so many aspects about this question, e.g. the usability does not depend only on completeness, but also accuracy, not to mention attribute completeness and accuracy, which are (understandably) not part of this study, and this paper may not be in a position to answer this RQ without a full data quality analysis beyond 'just' completeness (just to be clear: I am not discounting the importance and value of this work, only its ability to answer that research question). Further, each kind of urban analysis will have different application requirements and sensitivity to quality. What is good enough for one analysis, may not be suitable for another one. That is also why it is important to talk more about use cases (previous point).

- Figure 1 illustrates a temporal analysis. It is not clear whether the predicted completeness / total number of buildings is computed for each year, i.e. is the RF conducted for each snapshot of the data and using the explanatory variables specific to that year? I don't think so. I can imagine that in some areas there is a huge difference in the total amount of actual buildings across the selected time frame due to rapid urbanisation. So this aspect may potentially be quite flawed, unless I misunderstood the approach (in that case, it would be beneficial to expound a bit on the methodology and reasoning).

- In devising the method, did the authors consider the work and dataset presented in 10.1038/s41597-021-01105-4?

- The explanatory variables are not explained well. Besides more text, it would help to include a table with their overview, summary statistics, and examples of values for one or a few locations. Perhaps feature important could be useful too? To the extent of my understanding, this paper presents an entirely new method for estimating building completeness, of particular importance to the GIScience community, so these matters deserve more attention.

- I did not understand whether the regression model predicts the percentage of building completeness for each cell, or the sum of the building footprints i.e. site coverage (from which the % of completeness is computed when related to the area of buildings mapped in OSM). I would appreciate if this aspect could be made more explicit. Also, is it possible to predict the number of buildings rather than the area covered by buildings? I think that many completeness studies focus on the number of buildings rather than area, thus this study departs from traditional approaches, requiring some elaboration.

- I miss several points on the discussion on the variable level of completeness and in general the motivation for mapping buildings, which lag behind mapping roads in OSM. For example, while I like that various aspects such as the role of corporate editors in mapping buildings have been quantified, can the authors discuss what are the different motivations in mapping buildings, as opposed to road networks, amenities, etc.? Does the study result in recommendations for mapping buildings or suggestions how to improve the completeness? Why are corporate editors not interested in mapping buildings, at least when comparing to roads? As one of the authors is from a prominent corporation, there is a great chance to elaborate more on this matter if allowed.

- Would it make sense to discuss the role of large data imports? I understand that one of the authors recently published a paper on Analysing the Impact of Large Data Imports in OpenStreetMap; so there are surely first-hand insights that could be shared in the context of this paper -- would there be space to mention this aspect?

- How was the 1 km grid constructed? Is it adopted from the GHS-UCBD? If yes, perhaps clarify, as it is ambiguous.

- It is unclear to me how the RF model was tested -- how many cells did the authors analyse/test? Did they do so manually or did they simply compare it to authoritative datasets? How was the building area measured?

- First paragraph of the introduction -- please define what is SDG 11 and how can building data support its monitoring.

- Line 224: 'As a OSM data producer, you should use completeness maps' -- are these maps for all cities/countries published somewhere?

- Line 276: typo - geomorpholgy

- Figure 4 - it does not look like the maps include Central Asia, just a small part of it at the edge.

Abstract:

- SHDI -- acronym not defined. In fact, I don't think it's defined anywhere in the paper.

- Consider relating the result of 1,510 cities as a share of the 13,189 urban agglomerations / total number of cities included in the analysis: "Our results reveal that for 1,510 cities [xxx% of the analysed amount] OSM building footprint data exceeds 80% completeness"

Also, what is the population of these cities?

Reviewer #2 (Remarks to the Author):

This study utilized a regression model to infer OSM building completeness within 13,189 urban agglomerations. The authors found that: for 1,510 cities OSM building footprint data exceeds 80% completeness; Humanitarian mapping efforts have improved completeness; The digital divide in OSM has receded, but a strong spatial bias associated remains. Generally, the research work may be interesting to readers in the field of Earth Science and Data Science, and the manuscript is well written. However, a number of issues should be improved before it can be considered to be published.

1. Abstract: I cannot find the full name of 'SHDI'.
2. The authors found that "organized humanitarian mapping activities in urban centers contributed an average of about 8% of the building footprints globally." More details should be provided on how to get the value (8%).
3. L74-76. It has been stated that "The temporal evolution of urban building completeness showed similar patterns for urban centers regardless of their population (figure provided in online material only)." I cannot find where to download the 'material'. It is better to give out the corresponding link.
4. Figure 2: This figure has only been divided into three groups <20%, 20-80%, and >80%. More groups (e.g., 5-6) are needed to see the details.
5. Table1: It is not clear what are the specific criteria or values for dividing the urban centers, especially for the 'Human Development Index' and the 'population size'. Reasonable criteria should be considered.
6. L77-78: I think the 'Figure 2' should be mentioned near here.
7. L79: It has been highlighted that "for 1,510 cities OSM building footprint data exceeded 80% completeness." I understand that the authors want to highlight the cities with a high complete. But on the contrary, this figure also reported a low completeness for most cities worldwide, which is also worth to be mentioned and discussed.
8. L92-98: In this part, the authors analyzed the temporal variation of OSM building data completeness. In my view, it may also be valuable to show the distribution of OSM building completeness for different cities and different years, which may be useful to understand the temporal-spatial variation of OSM building data.
9. Figure 3: I notice that the Gini Coefficient tends to be stable since 2019 or around. However, this point has not been explained. Is the variation related to the COVID-19? I think a discussion of this point may be useful to understand the development of OSM building data in future.
10. L125: More details should be given out to explain why all the cities were divided into five different types. I guess the authors may refer to the hierarchical map in Figure 6c visually. But in my view, other quantitative and objective methods should also be considered.
11. Following the question (10), it is useful to give out the spatial distribution (or map) of different clusters of cities, and also to discussion the result.

Discussion

12. Microsoft has provided building footprint data for a number of countries worldwide. It seems that the Microsoft' data is much more complete than OSM data. Thus I am looking forward to see the comparison (including the spatial pattern) between these two datasets. Moreover, the discussion should also be carried out based on the comparison. For instance, as Microsoft has provided substantial building footprint data for most countries and cities, why should we still need to use the building data in OSM.

13. Similar to the question (12). The authors stated that " We showed that quality of building footprint data from Microsoft can be prone to low recall values in some areas." Despite of the flaw, we can see

from the analyses that the Microsoft's data is much more complete than OSM data in most countries and regions.

14. The study focused on OSM building data completeness. However, it is worth to compare the results with those reported in several existing studies that have also discussed OSM data quality at a global scale, e.g., what is the similarity and difference of various map features, roads, buildings and land-uses.

Reference

Barrington-Leigh and Millard-Ball. (2017). The world's user-generated road map is more than 80% complete. PLOS ONE.

Zhou et al. (2022). Exploring the accuracy and completeness patterns of global land-cover/land-use data in OpenStreetMap. Applied Geography.

Methods,

15. This study used reference building datasets covered 6,737 urban centers for both training and validating, and applied the trained model to refer to the completeness for 13,189 urban centers on a global scale. The main concern is that whether the model is applicable to other urban centers, because the building pattern may be quite different in different countries and regions. For instance, in China, there are more than 1,500 urban centers, but none of them were used as training and validation data. Was the trained model really applicable to the country? Besides, I notice that in the published materials- Data, the OSM completeness of some cities (e.g., Shanghai and Yogyakarta) is zero, which is not consistency with that we observed from the OSM platform.

16. Table 2. There may be errors in this table. I notice that the Microsoft did not provide building footprint data for some countries (e.g., China and Vietnam), If I am wrong, please provide the link for downloading these data. Another question is that because Microsoft has also provided building footprint data for a large number of countries and they are relatively more complete than OSM building data, why we still need to use OSM building data.

17. Table 3, The recall (or completeness) of Microsoft's data in some countries is relatively low (e.g., lower than 70%). My concern is that whether it is effective to use the Microsoft data for training and validation. In my view, other data (satellite) and approaches (e.g., visual inspection) should also be considered as a supplement.

18. L252: It is needed to specific in a table which variables have been used as input? Which are the determinant variables to obtain an effective model. More variables does not always ensure a better result.

19. L270: About the training mode. It seems that all the 6,500 urban centers have been used as samples for training and validating. Is it better to obtain a single train model for each country or region? because the building pattern may be similar in the regions close to each other.

20. Table 5: How to calculate 'MSE' and 'MAE', and what are the units of these two measures?

21. Figure 7. Although the scatter plot shows a strong correlation, I notice that the distribution of residuals may be different for different completeness values. For instance, the higher the completeness value, the larger the residual. That means the completeness with a high estimated value may have a larger bias. It is needed to carefully explain the results.

Data

22. Looking for some cities (e.g., Shanghai, Yogyakarta, Guilin) in the published dataset. The completeness of these cities are all equal to zero. But, there are a larger number of OSM building data, while checking in the OSM platform.

Reviewer #3 (Remarks to the Author):

This article reports on spatio-temporal properties of crowdsourced geospatial data. Specifically, the core focus is on bias in OpenStreetMap. This is an interesting and worthy topic. The main conclusion drawn from the work is that users of OpenStreetMap need to be aware of the biases, notably cautioning against naïve use of the map. It is also good to see that the data sets and code used are being made available which should help ensure replicability etc. The work is set in the context of the UN SDGs but this aspect feels a bit of a tag on.

There are some concerns with the article. In particular, it does not really cover the content suggested by its title – it does not really explore or test in detail the digital divides that exist but rather reports on some well-known properties of OpenStreetMap. The scope is limited to completeness of building footprints. This has interest but there are many other dimensions that could be considered (e.g. provenance and data quality in terms of trust, credibility and accuracy etc.). The assessments of completeness are also limited. As just one concern, the size of the omission errors indicated by the recall statistic in Table 3 suggests that the data are not really suitable as a reference data set. In addition, the geographic bias in the data (noted on line 251) fits uncomfortably with the article's core aim and conclusion.

The basic issues connected to the spatial and temporal biases in OpenStreetMap are also well known. This is particularly the case with the relatively easy to assess issue of building completeness. A quick search for papers on building completeness in OSM shows >4,000 articles in the last 4 years. The nature of OpenStreetMap contributors is also well known (e.g. they tend to be male, well-educated etc.) and the patterns of contribution (e.g. often on 'pet' topics, near home etc.) have been widely reported. Geographic biases, from basic urban v rural contrasts to regional scale imbalances are all very well known. Many studies have shown that the data for Europe and North America are more complete than elsewhere. Some other locations also have high completeness, especially associated with disasters (e.g. the OSM data for Haiti after the 2010 earthquake is a classic example used to promote OSM). The recent trend to bulk updates, often from people/organisations that can be regarded as relative expert contributors, is also well known.

The danger of using biased data is well known. The danger of using an average value (line 58) is really rather obvious and nothing more than common sense. Also although an aim is to determine if the data are 'good enough' (line 40) the authors never state what level of quality is required. The latter is also highly application dependent so for some users the data will be perfectly useful while for others they would be inappropriate.

Reviewer 1

Reviewer comment	The paper assesses OSM building completeness in a large number of urban agglomerations around the world. This is much needed work and the first such study doing so. The method is also novel, as typical OSM building completeness studies conducted at the national or city level cannot be scaled globally. The authors should be complimented on this important effort and valuable results. While to my understanding spatial data quality and OSM have not been subject of the journal, NC may be a suitable venue for it, and on the broader scope, it may also contribute to raising awareness of OSM among readers who may not be much into it. While I am positive about this work and would like to see it published in NC, I have several suggestions for improvement, some major, while some minor. If I misunderstood some concepts, the authors are advised to make some of the explanations in the paper clearer and more prominent.
Author Response	Thanks for the thorough review and well-conceived comments. We will address your comments point-by-point in the following paragraphs.

Reviewer comment	It may be questionable what constitutes a global study. And this may not be a truly global study. This is not necessarily a flaw, as I understand that the authors have focused only on urban areas (which is also obvious in the title), and that makes sense. It's definitely a significant advancement since previous attempts, but "only" 50% of the global population is taken into account, while much of the narrative may suggest otherwise. The focus is on areas that contain 183m buildings, while OSM at the moment has about 3x more buildings mapped than that. The authors argue multiple times that nobody should be left behind. But aren't they doing so too? Why not include all areas? Does the method really not work on rural areas, i.e. everywhere? If not, then the narrative should be adjusted a bit. I am not sure that this story is sensible as they investigate the global divide, but at the same time they perpetuate it by compounding the divide between urban and rural areas.
Author Response	We've carefully adjusted the narrative to not give the wrong impression that our work is covering the entire globe, although we use the term "global urban analysis" to refer to comparative analyses that include urban centres spread around the whole globe. Although our proposed method could potentially be applied beyond urban areas, this application is out of scope of the current study, as it would require adaptations in the datasets used and new evaluations. We provide a justification in the introduction of our focus on the urban built up areas, which are very policy relevant, the focus of several sustainable development goals, and anyway where most of the building footprints are located. We agree with the comment on the digital divide which is indeed not our primary focus - we adjusted the narrative to put less emphasis on the "digital divide", and more on the inequalities in completeness and their effects to analysis and policy decisions which would be based on data with strong biases in completeness. Title: Investigating completeness and inequalities in OpenStreetMap: spatio-temporal analysis of global urban building data

	Introduction line 43: A common data quality requirement for many research and policy applications based on building footprint data is the completeness of the building stock for analysis purposes. This is particularly important for comparative analyses that seek to discern global urban patterns, such as for instance to derive a "global" dataset of critical infrastructure\cite{Nirandjan2022}, or to use "big data" for comparing urban morphology across the globe\cite{Boeing2021}. When unaccounted for, spatial bias in completeness can lead urban analysts and researchers to draw general conclusions which are only valid for well-represented (well-mapped) areas\cite{Meyer2022}. Completeness of building stock data is particularly important for ensuring equitable and fair decision making based on OSM data for the policy and practice applications of Table \ref{table:use_cases} and, as such, has a direct importance for the overarching principle of the SDGs of "leaving no one behind".
--	--

Reviewer comment	The paper does not discuss much on the actual use of such data, especially at the large or even global scale (besides using it for city-scale analyses in a single city), which may be needed to assert the importance of the work. They do focus on SDG, but that is just one of many applications of building footprints. There have been recent studies that use a large volume of building footprints beyond one city, in fact, some of them do so using OSM building data, such as urban form, energy, and climate studies, e.g. the global building morphology indicators project. A few more sentences about why such data is important and how can it be used wouldn't hurt.
------------------	--

Author Response	We believe that this is a very important suggestion and we've put additional efforts into better outlining the use case of building footprint data in general and building footprint data from OpenStreetMap in particular. We think that by adding this information we can provide another value to the readers of the manuscript, which goes beyond the question of OSM building completeness. We included a new Table 1 which provides a set of illustrative examples of use cases applications that use OSM building data by stakeholders from the policy, research and business domains. In the introduction section, we've created a new table (Table 1), which highlights policy and practice examples and applications of global urban analysis using building footprint data. We've further refined the paragraph in the introduction about research works related to the use of building footprints for global urban analysis and SDG monitoring: Introduction line 13: Building data is an essential asset in global urban analyses for assessing progress towards a number of important urban goals. For instance, SDG Indicator 11.3.1 ("Ratio of land consumption rate to population growth rate") would directly benefit from building footprint data. However this indicator is currently mainly based on easily available remote sensing data, e.g. World Settlement Footprint (WSF)\cite{Esch2022}, which exacerbates the monitoring of structural changes such as changes in floorspace per capita or re-densification trends. The monitoring of SDG Indicator 11.1.1 ("Proportion of urban population living in slums, informal settlements or inadequate housing") would benefit from an analysis of building blocks and street networks considering their spatial relations, such as density and neighbourhood relations\cite{Brelsford2018}. SDG Indicator 11.7.1 ("Average share of the built-up area of cities that is open space for public use for all, by sex, age and persons with disabilities") can be partly derived from Sentinel-2 satellite data, but this data source falls short for distinguishing public from private green spaces as it fails to capture fine-grained urban structures due to its
-----------------	---

	10 metres resolution\cite{Ludwig2021}. Global Building Morphology Indicators could allow us to quantify the form of urban areas and enable novel analyses comparing morphological parameters across cities which go much further than existing approaches which are often based on aggregated population data\cite{Biljecki2022}. In addition to these examples of urban research using global building footprint data, Table \ref{table:use_cases} provides recent policy examples and practical applications of global urban analysis using building footprint data.
--	---

Reviewer comment	Somewhat in relation to the previous point, I have a comment pertaining to the first RQ 'Is OpenStreetMap building data good enough for urban analysis and SDG monitoring?' -- I am not sure the paper actually answers this RQ, at least adequately. There are so many aspects about this question, e.g. the usability does not depend only on completeness, but also accuracy, not to mention attribute completeness and accuracy, which are (understandably) not part of this study, and this paper may not be in a position to answer this RQ without a full data quality analysis beyond 'just' completeness (just to be clear: I am not discounting the importance and value of this work, only its ability to answer that research question). Further, each kind of urban analysis will have different application requirements and sensitivity to quality. What is good enough for one analysis, may not be suitable for another one. That is also why it is important to talk more about use cases (previous point).
------------------	---

Author Response	We agree with the reviewer and slightly adjusted RQ1 and RQ2, so that our results and methodological approach to assess completeness and inequality in OSM are better reflected. Introduction line 61: RQ1: What is the completeness of OpenStreetMap building data in the context of global urban analysis applications? RQ2: How unequal is urban OpenStreetMap building data distributed within the space of a city, across continents and on the global level?
-----------------	---

Reviewer comment	Figure 1 illustrates a temporal analysis. It is not clear whether the predicted completeness / total number of buildings is computed for each year, i.e. is the RF conducted for each snapshot of the data and using the explanatory variables specific to that year? I don't think so. I can imagine that in some areas there is a huge difference in the total amount of actual buildings across the selected time frame due to rapid urbanisation. So this aspect may potentially be quite flawed, unless I misunderstood the approach (in that case, it would be beneficial to expound a bit on the methodology and reasoning).
------------------	---

Author Response	The predicted number of buildings per 1km grid cells is performed for a single snapshot of the explanatory variables, which is not considering changes to these variables over time. We have added Table 3 to the methodology section, which provides summary statistics for explanatory variables utilized in the machine learning model and highlights for each variable the year for which data was captured. In revising the data for the paper, we have updated the GHS-POP data from the 2015 to the newest product from 2020. We added an explanation to the methodology section to be transparent about this uncertainty and adjusted the caption of Figure 1. Caption figure 1: Building footprint predictions were based on explanatory data for 2020. Therefore, the uncertainty of building completeness estimates increases with increasing distance to 2020. This is not reflected in the confidence bands as this additional uncertainty is
-----------------	--

	hard to quantify. Methodology line 298: The data values for ESA WorldCover, GHS-POP and VNL are subject to uncertainty, as they reflect the situation for the year 2020, albeit they are utilized in a model to assess OSM building completeness for a time range from 2008 to 2023. The model will be most suited to predict the building stock of 2020 and not the past building stock, nor the correct building stock for 2023. As a consequence, the analysis will slightly overestimate completeness for regions which have seen rapid urbanisation since 2020, but might also underestimate completeness for timestamps before 2020.
--	--

Reviewer comment	In devising the method, did the authors consider the work and dataset presented in 10.1038/s41597-021-01105-4?
Author Response	We've checked to what extent this dataset could be used in our approach and utilized it to perform an independent assessment of the Microsoft Global ML Building Footprints dataset. Previously we could only report on the self-declared performance estimates by Microsoft. We have added Table 8 to the manuscript, which reports precision and recall for the Microsoft building footprints by comparing to Geo-Wiki built-up surface validation dataset. Methodology line 268: The Geo-Wiki built-up reference dataset^{\cite{See2022}} has been utilized to assess the suitability of the Microsoft building footprint dataset for our OSM completeness modelling approach. The Geo-Wiki campaign visually assessed very high-resolution satellite images of 50 K sample locations for the presence of built-up surfaces containing any building with a roof using a crowdsourcing approach^{\cite{See2022}}. All Geo-Wiki grid cells intersecting the urban center geometries for which Microsoft building data was available at the city level were considered in the analysis. Precision and recall for the Microsoft building footprints have been derived on the 10x10 meters Geo-Wiki grid level (see Supplementary Material Section Table ^{\ref{table:MSquality_geowiki_validation}}). Grid cells were defined as true positives (TP) if at least one Microsoft building footprint intersected with a Geo-Wiki grid cell labelled as 'Built-up'. True negatives (TN) were defined as Geo-Wiki grid cells labelled as 'Not built-up' that did not intersect with any Microsoft building footprint. Accordingly we defined false positives (FP) as grid cell labelled as 'Not built-up' which intersected with at least one Microsoft building and false negatives (FN) as the reverse. Finally, we also provide precision and recall for the Microsoft building footprints as self-declared by Microsoft in the Global ML Building Footprints GitHub repository (see Supplementary Material Section Table ^{\ref{tab:MSquality}}).

Reviewer comment	The explanatory variables are not explained well. Besides more text, it would help to include a table with their overview, summary statistics, and examples of values for one or a few locations. Perhaps feature important could be useful too? To the extent of my understanding, this paper presents an entirely new method for estimating building completeness, of particular importance to the GIScience community, so these matters deserve more attention.
Author Response	Thanks for this very important comment. We want to be as clear as possible on the variables used for the machine learning model and provide additional information in

	the manuscript, supplementary material and GitHub code repository. We have added Table 3 to the methodology section, which provides summary statistics for the explanatory variables utilized in the machine learning model.
--	--

Reviewer comment	I did not understand whether the regression model predicts the percentage of building completeness for each cell, or the sum of the building footprints i.e. site coverage (from which the % of completeness is computed when related to the area of buildings mapped in OSM). I would appreciate if this aspect could be made more explicit. Also, is it possible to predict the number of buildings rather than the area covered by buildings? I think that many completeness studies focus on the number of buildings rather than area, thus this study departs from traditional approaches, requiring some elaboration.
------------------	---

Author Response	We've made it more clear in the methodology section that the model predicts the building area per 1km grid cell. For this study we decided to use an area-based over a count-based measure to account for differences in the type of modeling buildings in OSM and reference datasets. Methodology line 307: We used a Random Forest (RF) regressor \cite{Breiman2001a} to predict the overall building area in square meters per 1x1 kilometer grid cell using the covariates described in the section above. Building completeness is not directly predicted, but inferred in a second step using this prediction and the corresponding surface area of all OSM buildings per grid cell. Methodology line 348: For each urban center we calculated the OSM building completeness using the area ratio method (reference building area / OSM building area) which has been applied by several other researchers in the context of urban areas \cite{Fan2014,Hecht2013}. We did not consider the building count, but building area instead, to account for the high sensitivity to disparities in modelling when using unit based completeness measures \cite{Hecht2013}, which is especially important when using a wide range of different building datasets as in this study. For instance, some datasets, such as OS OpenMap Local, model terraced housing as a single polygon, whereas in OSM these buildings are usually mapped as individual geometrical features. Whereas this would result only in a slight difference in the surface area of all buildings per grid cell, the building count can vary dramatically and completeness might be overestimated.
-----------------	---

Reviewer comment	I miss several points on the discussion on the variable level of completeness and in general the motivation for mapping buildings, which lag behind mapping roads in OSM. For example, while I like that various aspects such as the role of corporate editors in mapping buildings have been quantified, can the authors discuss what are the different motivations in mapping buildings, as opposed to road networks, amenities, etc.? Does the study result in recommendations for mapping buildings or suggestions how to improve the completeness? Why are corporate editors not interested in mapping buildings, at least when comparing to roads? As one of the authors is from a prominent corporation, there is a great chance to elaborate more on this matter if allowed.
------------------	--

Author	Understanding the motivations of contributions to OSM is an interesting field of
--------	---

Response	research and the trend towards organized corporate or humanitarian mapping has added new perspectives towards this. From our point of view, we believe that corporates are interested in using building data on a global scale, but are not necessarily relying on OSM as the only source for this. In addition, we think that the conflation of two building datasets is easier to achieve than conflating two different road datasets, especially when aiming for a topological network. We address these points in the introduction section. Introduction Section line 31: So far corporates have focused on mapping roads in OSM, but they are also interested in building data given the efforts invested into automated machine identification, such as the Microsoft Building open dataset\cite{MSbuildingfootprints}. These features have been released in an OSM compatible format, but not through OSM specifically, but one could feasibly combine these two datasets\cite{Daylight}. Roads, on the other hand, require significantly more effort to conflate, needing to maintain referential integrity and a topological network to operate properly. Working in OSM directly solves this issue.
----------	---

Reviewer comment	Would it make sense to discuss the role of large data imports? I understand that one of the authors recently published a paper on Analysing the Impact of Large Data Imports in OpenStreetMap; so there are surely first-hand insights that could be shared in the context of this paper -- would there be space to mention this aspect?
------------------	--

Author Response	Imports are a fiercely disputed topic within the OSM community. We've added a sentence to the discussion section to briefly touch on the topic of data imports from a researcher's perspective. We acknowledge that there would be much more to discuss about the topic of data imports, but the limited space available for this paper prevents us from providing more details. Discussion line 239: A large direct import of Microsoft building data into OSM seems unlikely at the moment, however it has been shown for the road network that imports can increase contributor activity especially for already engaged mappers and an interplay of data imports and updates by contributors could improve OSM data dramatically\cite{Witt2021,Zielstra2013}.
-----------------	--

Reviewer comment	How was the 1 km grid constructed? Is it adopted from the GHS-UCBD? If yes, perhaps clarify, as it is ambiguous.
------------------	--

Author Response	We added information in the Methods section and clarified that the grid has been adopted from GHS-UCBD. Methodology line 253: The grid adopted the same structure utilized by the raster datasets of the GHS-UCBD.
-----------------	--

Reviewer comment	It is unclear to me how the RF model was tested -- how many cells did the authors analyse/test? Did they do so manually or did they simply compare it to authoritative datasets? How was the building area measured?
------------------	--

Author Response	We've added further information on how the building area was derived for the reference datasets and make clear that the reference data was used to evaluate the performance of our model.
-----------------	--

	Methodology line 256: For each grid cell we derived the overall building footprint area in square kilometers for each of the reference datasets by intersecting the grid cells and building footprints and then summing up the corresponding surface area of all building (parts) per grid cell. In case two reference datasets were available, e.g. from Microsoft and an authoritative source, we considered the information from the authoritative dataset. Methodology line 327: On the grid level, MAE describes the average of the absolute differences between predicted building area and reference building area obtained from authoritative sources or Microsoft's Global ML Building footprints in square kilometers. Accordingly, MSE describes the average of the squared differences between predicted building area and reference building area per grid cell. On the urban centers level, MAE describes the average of the absolute differences between predicted building completeness and reference building completeness. Accordingly, MSE describes the average of the squared differences between predicted building completeness and reference building completeness per urban center. These measures were calculated based on scikit-learn's python implementation\cite{scikitLearn2011}.
--	---

Reviewer comment	First paragraph of the introduction -- please define what is SDG 11 and how can building data support its monitoring.
Author Response	We've added more information on the general aim of SDG 11 and highlighted how the sub-goals of SDG 11 can benefit from building data. Introduction line 7: Improving the systematic monitoring of the global urbanization process is a requirement for achieving the United Nation's Sustainable Development Goals (SDGs), e.g. "urban" SDG 11 ("Make cities and human settlements inclusive, safe, resilient and sustainable."), especially in the low-income countries where the data are usually scarce\cite{Sun2020}. Introduction line 13: Building data is an essential asset in global urban analyses for assessing progress towards a number of important urban goals. For instance, SDG Indicator 11.3.1 ("Ratio of land consumption rate to population growth rate") would directly benefit from building footprint data. However this indicator is currently mainly based on easily available remote sensing data, e.g. World Settlement Footprint (WSF)\cite{Esch2022}, which exacerbates the monitoring of structural changes such as changes in floorspace per capita or re-densification trends. The monitoring of SDG Indicator 11.1.1 ("Proportion of urban population living in slums, informal settlements or inadequate housing") would benefit from an analysis of building blocks and street networks considering their spatial relations, such as density and neighbourhood relations\cite{Brelsford2018}. SDG Indicator 11.7.1 ("Average share of the built-up area of cities that is open space for public use for all, by sex, age and persons with disabilities") can be partly derived from Sentinel-2 satellite data, but this data source falls short for distinguishing public from private green spaces as it fails to capture fine-grained urban structures due to its 10 metres resolution\cite{Ludwig2021}.

Reviewer comment	Line 224: 'As a OSM data producer, you should use completeness maps' -- are these maps for all cities/countries published somewhere?
------------------	---

Author Response	We have invested additional efforts to make the completeness maps available via the ohsomeHex platform, which allows users to interactively explore the completeness at the urban centers level and the grid level for the entire time range covered by this analysis. All data is furthermore available in GitHub. https://hex.ohsome.org/#/urban_building_completeness global distribution for 2023-01-01 [redacted] intra-urban completeness for the city of Berlin for 2011-01-01 [redacted]
---

Reviewer comment	Line 276: typo - geomorpholgy
Author Response	Thanks for your careful reading. We fixed this typo.

Reviewer comment	Figure 4 - it does not look like the maps include Central Asia, just a small part of it at the edge.
Author Response	We want to apologize for this careless mistake. We've adjusted the design of Figure 4 to include the urban centers in Central Asia, not only displaying urban centers in Europe.

Reviewer comment	Abstract: SHDI -- acronym not defined. In fact, I don't think it's defined anywhere in the paper.
Author Response	SHDI stands for "Subnational Human Development Index". We defined the acronym in the manuscript now. Introduction line 68: The model further relies on information obtained from remote sensing data (land cover, population distribution, night time lights), Subnational Human Development Index (SHDI), and urban road network density as predictors.

Reviewer comment	Abstract: Consider relating the result of 1,510 cities as a share of the 13,189 urban agglomerations / total number of cities included in the analysis: "Our results reveal that for 1,510 cities [xxx% of the analysed amount] OSM building footprint data exceeds 80% completeness" Also, what is the population of these cities?
Author Response	We've added this information to the sentence in the abstract and added further information in the Results section of the manuscript. Abstract: For 1,848 urban centres (16% of the urban population), OSM building footprint data exceeds 80% completeness, but completeness remains lower than 20% for 9,163 cities (48% of the urban population). Results line 74: Our results reveal that for 1,848 cities (14% of the analysed amount) OSM building footprint data exceeded 80% completeness. In total, these cities were home to a population of 492 Million people (16% of the global urban population). Contrary, our results show that for 9,163 cities (69% of the analysed amount and home to 48% of the global urban population) OSM building footprint data did not reach 20% completeness.

Reviewer 2

Reviewer comment	This study utilized a regression model to infer OSM building completeness within 13,189 urban agglomerations. The authors found that: for 1,510 cities OSM building footprint data exceeds 80% completeness; Humanitarian mapping efforts have improved completeness; The digital divide in OSM has receded, but a strong spatial bias associated remains. Generally, the research work may be interesting to readers in the field of Earth Science and Data Science, and the manuscript is well written. However, a number of issues should be improved before it can be considered to be published.
Author Response	Thank you for reviewing our paper. We will address your suggestions and comments in the following section.

Reviewer comment	1. Abstract: I cannot find the full name of 'SHDI'.
Author Response	SHDI stands for "Subnational Human Development Index". We defined the acronym in the manuscript now. Introduction line 68: The model further relies on information obtained from remote sensing data (land cover, population distribution, night time lights), Subnational Human Development Index (SHDI), and urban road network density as predictors.

Reviewer comment	2. The authors found that "organized humanitarian mapping activities in urban centers contributed an average of about 8% of the building footprints globally." More details should be provided on how to get the value (8%).
Author Response	We've added a reference to Table 2, which provides this information. More information on how this value was derived can be found in the Methodology section. We've added more information to clarify how the number was derived. Results line 84: We found that organized humanitarian mapping activities in urban centers contributed an average of about 10% of the building footprints globally (see Table \ref{table:results}). Methodology line 365: We investigated the impact of humanitarian mapping through the HOT Tasking Manager and corporate mapping by Apple, Meta, MapBox, Microsoft and Kaart on overall completeness and inequality measures. OSM contributions have been considered as humanitarian mapping activities following the approach developed by Herfort et al. (2021) which utilizes information obtained from a HOT Tasking Manager database dump \cite{Herfort2021} . Corporate mapping activities were identified by OSM user ID, expanding on the approach presented in \cite{Anderson2019} by using a mapper's self-disclosed corporate affiliation in their OSM user bio instead of relying on out-of-date lists on the OSM wiki \cite{OSMWikiOrganizedEditing} . According to these two approaches, the contributions of each OSM user were first grouped into either "humanitarian", "corporate" or "other" and in a second step according to World Bank region, SDHI class and city size class by summing up the building area added to OSM per map edit. Based on this information, we derived the share of humanitarian map edits and

	corporate map edits on the overall OSM building data considering the building area (and not building count) as the base unit.
--	---

Reviewer comment	3. L74-76. It has been stated that "The temporal evolution of urban building completeness showed similar patterns for urban centers regardless of their population (figure provided in online material only)." I cannot find where to download the 'material'. It is better to give out the corresponding link.
------------------	---

Author Response	We've added the file as supplementary material now. In addition we had already uploaded this file to the GitHub Repository where all data and code has been published. https://github.com/GIScience/global-urban-building-completeness-analysis/blob/master/figures/completeness_per_month_by_region_rf_adjusted_population.png
-----------------	---

Reviewer comment	4. Figure 2: This figure has only been divided into three groups <20%, 20-80%, and >80%. More groups (e.g., 5-6) are needed to see the details.
------------------	---

Author Response	We've adjusted the map design and introduced a more detailed visualization, which now consists of 5 classes: <20%, 20% - 40%, 40% - 60%, 60% - 80% and > 80%. This version now allows the reader to see the necessary details and also makes the paper more consistent, as the same symbology and classes were used in Figure 6 already, depicting the completeness for grid cells for selected urban centers.  OpenStreetMap Building Completeness in Urban Centers Completeness  <20% [9163] 20% - 40% [908] 40% - 60% [588] 60% - 80% [682] >=80% [1848] Population  40M 10M 1M Author: B. Herfort, 2023 Data: OpenStreetMap contributors, Urban Centre Database UCDB R2019A v1.2 HeiGIT HEIDELBERG INSTITUTE FOR GEOINFORMATION TECHNOLOGY 
-----------------	---

Reviewer comment	5. Table1: It is not clear what are the specific criteria or values for dividing the urban centers, especially for the 'Human Development Index' and the 'population size'. Reasonable criteria should be considered.
------------------	---

Author Response	We adjusted the caption of this Table (now Table 2) and added information on how urban centers have been divided in regard to the Subnational Human Development Index and Population size. This information was also already available in the Methodology section.
-----------------	--

	Methodology Section line 360: SHDI classes were based on cut-off points defined by the United Nations Development Programme\cite{UnitedNationsDevelopmentProgramme2019}: low human development (SHDI< 0.550), medium human development (SHDI: 0.550 - 0.699), high human development (SHDI: 0.700–0.799), very high human development (SHDI> 0.800). City size classes were based on population thresholds defined by OECD\cite{oecd2016}: small urban areas (50k–200k), medium-size urban areas (200k–500k), metropolitan areas (500k–1.5M), large metropolitan areas (>1.5M).
--	--

Reviewer comment	6. L77-78: I think the 'Figure 2' should be mentioned near here.
Author Response	We have added a reference to Figure 2 in the highlighted sentence to connect the text and figure in a better way. Results Section line 101: The spatial distribution of building completeness across urban centers shows a strong regional variability across that global trend: numerous cities in any region were mapped with a very high completeness regardless of the overall completeness or mapping activity in that region (see Figure \ref{fig:completeness_world_map}).

Reviewer comment	7. L79: It has been highlighted that "for 1,510 cities OSM building footprint data exceeded 80% completeness." I understand that the authors want to highlight the cities with a high complete. But on the contrary, this figure also reported a low completeness for most cities worldwide, which is also worth to be mentioned and discussed.
Author Response	We added further information in the Results section of the manuscript to also highlight the number of cities and urban population living in regions with low OSM building completeness. Results Section line 74: Our results reveal that for 1,848 cities (14% of the analysed amount) OSM building footprint data exceeded 80% completeness. In total, these cities were home to a population of 492 Million people (16% of the global urban population). Contrary, our results show that for 9,163 cities (69% of the analysed amount and home to 48% of the global urban population) OSM building footprint data did not reach 20% completeness.

Reviewer comment	8. L92-98: In this part, the authors analyzed the temporal variation of OSM building data completeness. In my view, it may also be valuable to show the distribution of OSM building completeness for different cities and different years, which may be useful to understand the temporal-spatial variation of OSM building data.
Author Response	We've produced a new figure (Figure 7) to visualize the temporal evolution of OSM building completeness at the urban center level and highlighted the temporal evolution for the cluster representatives. We acknowledge that it would be very interesting to analyze the spatial distribution of completeness at different points in time per urban center in further detail. However, we also believe that this deserves more elaboration on the right methodology and is

	more than what we can add to this manuscript. Results Section line 158: Figure \ref{fig:temporal_evolution_urban_completeness} highlights that for the case of Paris between 2010 and 2013 almost 80\% of the entire building stock was added to OSM. Discussion Section line 228: A more detailed analysis of the temporal trajectories for urban centers would facilitate this investigation and might reveal to what extent data quality dimensions, temporal evolution of mapping activity and inequality measures might be mutually dependent.
--	--

Reviewer comment	9. Figure 3: I notice that the Gini Coefficient tends to be stable since 2019 or around. However, this point has not been explained. Is the variation related to the COVID-19? I think a discussion of this point may be useful to understand the development of OSM building data in future.
Author Response	We've added further information about this in the Results section. Results line 120: Attention should be paid to the fact that Gini coefficient and Moran's I have been stagnating on the global level since 2019, which might indicate a shift in mapping behaviour due to restrictions caused by the COVID-19 pandemic. However, since 2021 Gini coefficient started to increase again, implying a tendency towards a more even distribution of building mapping across urban centers on the global level.

Reviewer comment	10. L125: More details should be given out to explain why all the cities were divided into five different types. I guess the authors may refer to the hierarchical map in Figure 6c visually. But in my view, other quantitative and objective methods should also be considered.
Author Response	We've calculated Silhouette Score and Calinski Harabasz Index Score for a different number of clusters and added a Table (Table 10) to the supplementary material. We discussed the scores and based on these scores we decided to go with 3 main clusters and 2 sub types for cluster (1) and cluster (2). Methodology line 409: The investigation was complemented by an agglomerative hierarchical cluster analysis of urban centers considering evenness and clustering within each city. The number of clusters has been selected based on the hierarchical structure of the full dendrogram (see Figure \ref{fig:intra_urban_completeness_inequality_map}) and by investigating the Variance Ratio Criterion (Calinski-Harabasz Index) and Silhouette Coefficient for various number of clusters (see Supplementary Material Section Table \ref{table:clustering_performance}). This analysis revealed that a number of two or three clusters would be optimal when only considering the clustering performance scores. We decided for three main clusters (1-3), but also report sub clusters for 1 (a, b) and 2 (a, b) to allow for a more fine grained distinction of urban centers, especially for those with low or medium completeness.

Reviewer comment	11. Following the question (10), it is useful to give out the spatial distribution (or map) of different clusters of cities, and also to discussion the result.
------------------	--

Author Response	We've produced a new figure for the supplementary materials section (Figure 11) to visualize the spatial distribution of urban centers in regard to their cluster on a map. Results Section line 136: The spatial distribution of urban centers based on the intra-urban inequality (Figure \ref{fig:clustering_world_map} Supplementary Materials section) shows a similar spatial pattern as Figure \ref{fig:completeness_world_map}.
-----------------	--

Reviewer comment	12. Microsoft has provided building footprint data for a number of countries worldwide. It seems that the Microsoft' data is much more complete than OSM data. Thus I am looking forward to see the comparison (including the spatial pattern) between these two datasets. Moreover, the discussion should also be carried out based on the comparison. For instance, as Microsoft has provided substantial building footprint data for most countries and cities, why should we still need to use the building data in OSM.
------------------	---

Author Response	Thanks for this valuable comment. We acknowledge that the comparison of OSM buildings and Microsoft buildings is a relevant topic and provide initial findings in the discussion section and as supplementary material. Nevertheless, we also see the need to address this topic with the required (methodological) courtesy, which we are not fully able to achieve within this manuscript. In general, OSM might be incomplete, but is going to have higher accuracy than Microsoft which might have more buildings, but more false positives. The best dataset is then going to be a combination of MS and OSM buildings: Using MS buildings where OSM buildings may not exist, but using OSM buildings when they are present because they have been verified. The Rapid OSM editor includes the MS buildings when users are editing, allowing an OSM editor to verify or reject the MS building prediction. This results in the building being verified and now existing in both datasets. In that sense, OSM's main advantage over Microsoft's buildings is the fact that OSM data is validated by humans. Discussion Section line 231: Future work should further investigate the potential of a harmonious ensemble dataset that combines the best of OpenStreetMap buildings with additional building coverage from deep learning based datasets such from Microsoft Buildings\cite{MSbuildingfootprints}. We have performed the direct comparison between OSM and Microsoft buildings and report the findings in the Supplementary Materials section (Table \ref{table:comparison_osm_microsoft} and Figure \ref{fig:comparison_osm_microsoft_world_map}). Ultimately, OSM buildings and MS buildings represent two very different types of data, and the major advantage of OSM is that it is continuously verified by human editors. This is the reason why not all of the Microsoft buildings are in OSM, as the Microsoft buildings represent a different type of dataset: algorithmically extracted from aerial imagery using automated methods. For some places, the building footprints in OSM might already come from an imported dataset, but they need to be accepted by a human-in-the-loop process, for instance using the mapwith.ai editor \cite{mapwith_ai} or similar tools. A large direct import of Microsoft building data into OSM seems unlikely at the moment, however it has been shown for the road network that imports can increase contributor activity especially for already engaged mappers and an interplay of data imports and updates by contributors could improve OSM data significantly\cite{Witt2021,Zielstra2013}.
-----------------	--

Reviewer comment	13. Similar to the question (12). The authors stated that " We showed that quality of building footprint data from Microsoft can be prone to low recall values in some areas." Despite of the flaw, we can see from the analyses that the Microsoft's data is much more complete than OSM data in most countries and regions.
Author Response	It is correct that in comparison to OSM the Microsoft building footprints do also cover many regions for which as of now no OSM building data is available. We address this point in more detail above for question (12).

Reviewer comment	14. The study focused on OSM building data completeness. However, it is worth to compare the results with those reported in several existing studies that have also discussed OSM data quality at a global scale, e.g., what is the similarity and difference of various map features, roads, buildings and land-uses. Reference Barrington-Leigh and Millard-Ball. (2017). The world's user-generated road map is more than 80% complete. PLOS ONE. Zhou et al. (2022). Exploring the accuracy and completeness patterns of global land-cover/land-use data in OpenStreetMap. Applied Geography.
Author Response	Thanks for this valuable comment. We've added/highlighted further references to other OSM data quality related research. Discussion section line 166: The results confirm the similar, albeit less dichotomous, global completeness pattern of land use land cover (LULC) information from OSM\cite{Zhou2022} and stand in slight contrast to the relatively higher and more evenly distributed completeness for OSM's road network\cite{BarringtonLeigh2017}. This highlights the complexity of OSM mapping activities and the challenge for urban analyses based on OSM data. Discussion section line 219: Our findings extend the general pattern of urban OSM building completeness as of 2020-01\cite{Zhou2022b} a) by highlighting the temporal evolution of mapping activity in conjunction with other events such as the COVID-19 pandemic, b) by considering a spatially much more balanced and extensive training data set from various sources and c) by investigating the global pattern and consequences of inequalities in completeness. In addition, geospatial data quality is comprised of dimensions beyond measuring completeness\cite{Oort2006}. For some sectors, such as public health programs, assessment of completeness is only the first step, and information on building usage is also required, but often only available for a small subset \cite{Sturrock2018}. It has been shown for LULC in OSM that the spatial pattern for completeness and accuracy are not necessarily the same\cite{Zhou2022}. Whereas urban OSM building completeness and OSM LULC completeness show similar global trends on the national level, OSM building data accuracy needs to be further investigated, e.g. towards its potential utilization as training and/or validation samples in machine learning models.

Reviewer comment	15. This study used reference building datasets covered 6,737 urban centers for both training and validating, and applied the trained model to refer to the completeness for 13,189 urban centers on a global scale. The main concern is that whether the model
------------------	---

	is applicable to other urban centers, because the building pattern may be quite different in different countries and regions. For instance, in China, there are more than 1,500 urban centers, but none of them were used as training and validation data. Was the trained model really applicable to the country? Besides, I notice that in the published materials- Data, the OSM completeness of some cities (e.g., Shanghai and Yogyakarta) is zero, which is not consistency with that we observed from the OSM platform.
Author Response	Thanks for this very valuable comment and for making use of the data we provided. This is a very apparent lesson, why it is so important to provide access to research data and code. Based upon your comment we double checked our dataset and found that OSM building area statistics (and accordingly OSM completeness information) were accidentally missing for the following seven urban centers: Ajdabiya, Baalbek, Kharagpur, Mengzi, Zhuhai, Yogyakarta and Shanghai. We've fixed this issue and updated the data for the missing cities and the code in our GitHub repository. We argue that while there are for sure differences in the urban structure of cities across the globe, these do not lead to severe structural differences which affect the transferability of the model. However, we highlighted this potential limitation in the Discussion Section. Discussion Section line 208: We were also not able to quantify the uncertainty for countries with a large number of urban centers (e.g. China) for which training data was not available. Nevertheless, other authors have highlighted that rapidly urbanizing cities, for instance in China\cite{Cervero2008}, are "mimicking [suburbanisation] trends and patterns of the post-World-War-II United States". As such, we assume no severe structural deviations in respect to the modeled relationships between explanatory variables and building area prediction for these countries. The low feature importance for the variable "World Bank Region Code" seems to be in line with that assumption.

Reviewer comment	16. Table 2. There may be errors in this table. I notice that the Microsoft did not provide building footprint data for some countries (e.g., China and Vietnam), If I am wrong, please provide the link for downloading these data. Another question is that because Microsoft has also provided building footprint data for a large number of countries and they are relatively more complete than OSM building data, why we still need to use OSM building data.
Author Response	The latest release of Microsoft building footprints does indeed include information about building footprints for Vietnam. The entire coverage of the dataset (which has evolved over the past months) is available here: https://github.com/microsoft/GlobalMLBuildingFootprints But, as pointed out correctly Microsoft building footprints do not include buildings in China. We've carefully checked our database and found a single grid geometry for the Vietnamese town Mong Cai, which is located at the border of China and Vietnam, which was accidentally counted as Chinese. We have corrected this and removed the entry for China in the table (now Table 3). In our response to question (12) we've pointed out a few differences and similarities for OSM and Microsoft building data. In addition, we believe that no single dataset will provide best results, but a combination of OSM, Microsoft buildings (and potentially other sources). One major strength of OSM over algorithmically derived data lies in it's potential to engage local communities, it's cultural and social openness and the

	fact that it's human derived information. Discussion Section line 187: In light of the critical challenges to finance high-quality data systems for addressing inequalities in SDG monitoring in both low and middle income countries\cite{Ulbrich2019}—despite heightened demand\cite{Sachs2022}—the creation and usage of OSM data could be promoted further as a cost-effective alternative. Data generation with OSM not only allows filling the current data gaps essential to monitor progress, but can also be an pathway for equitable urban transformations by empowering local communities to have a voice and benefit from the data production process\cite{PortodeAlbuquerque2021}. The cultural openness and social nature of OSM is a clear strength to achieve transparency about existing inequalities and how to address them, especially in comparison with building footprint datasets that are derived using proprietary, black-box machine learning approaches for which bias and fairness are often still unknown\cite{Mehrabi2021}. Discussion Section line 231: Future work should further investigate the potential of a harmonious ensemble dataset that combines the best of OpenStreetMap buildings with additional building coverage from deep learning based datasets such from Microsoft Buildings\cite{MSbuildingfootprints}.
--	--

Reviewer comment	17. Table 3, The recall (or completeness) of Microsoft's data in some countries is relatively low (e.g., lower than 70%). My concern is that whether it is effective to use the Microsoft data for training and validation. In my view, other data (satellite) and approaches (e.g., visual inspection) should also be considered as a supplement.
------------------	---

Author Response	We checked if Microsoft data is good enough (completeness) for training and validation by utilizing the validation dataset provided by the Geo-Wiki project. Results were considered of high enough quality for the urban areas considered in our analysis. Results Section line 203: The quality of building footprint data from Microsoft (c.f. Supplementary Material Table \ref{table:MSquality_geowiki_validation}) showed a sufficient recall of more than 80% for urban centers in all regions. However, self-declared recall values by Microsoft have been considerably lower for some regions. This indicates that recall might be lower in rural areas, which have not been included into our study. Nevertheless, the remaining biases of the Microsoft buildings dataset for urban areas are also reflected in the results reported here and might lead to too low building area predictions and consequently too high OSM building completeness estimates. Methodology line 268: The Geo-Wiki built-up reference dataset\cite{See2022} has been utilized to assess the suitability of the Microsoft building footprint dataset for our OSM completeness modelling approach. The Geo-Wiki campaign visually assessed very high-resolution satellite images of 50 K sample locations for the presence of built-up surfaces containing any building with a roof using a crowdsourcing approach\cite{See2022}. All Geo-Wiki grid cells intersecting the urban center geometries for which Microsoft building data was available at the city level were considered in the analysis. Precision and recall for the Microsoft building footprints have been derived on the 10x10 meters Geo-Wiki grid level (see Supplementary Material Section Table
-----------------	---

	\ref{table:MSquality_geowiki_validation}. Grid cells were defined as true positives (TP) if at least one Microsoft building footprint intersected with a Geo-Wiki grid cell labelled as 'Built-up'. True negatives (TN) were defined as Geo-Wiki grid cells labelled as 'Not built-up' that did not intersect with any Microsoft building footprint. Accordingly we defined false positives (FP) as grid cell labelled as 'Not built-up' which intersected with at least one Microsoft building and false negatives (FN) as the reverse. Finally, we also provide precision and recall for the Microsoft building footprints as self-declared by Microsoft in the Global ML Building Footprints GitHub repository (see Supplementary Material Section Table \ref{tab:MSquality}).
--	--

Reviewer comment	18. L252: It is needed to specific in a table which variables have been used as input? Which are the determinant variables to obtain an effective model. More variables does not always ensure a better result.
Author Response	We've added a new Table (Table 3) which provides information on Min, Max, Mean, Median and Feature Importance for each explanatory variable. Methodology line 303: Initially, we considered additional explanatory variables (e.g. permanent water bodies, fossil fuel consumption, OSM railway length, OSM amenity count), but these have been disregarded as their feature importance in the Random Forest model turned out to be very low (<0.02).

Reviewer comment	19. L270: About the training mode. It seems that all the 6,500 urban centers have been used as samples for training and validating. Is it better to obtain a single train model for each country or region? because the building pattern may be similar in the regions close to each other.
Author Response	As we did not have access to training data for every country, we decided to utilize a global model instead. The low feature importance for the variable "World Bank Region Code" signals that regional differences between the countries for which training data was available were of minor importance. Hence, we concluded that a global model (based on a larger set of training samples), is better suited for our application to borrow strength from urban centers which are not spatially adjacent, but still share many similarities. Discussion Section line 208: We were also not able to quantify the uncertainty for countries with a large number of urban centers (e.g. China) for which training data was not available. Nevertheless, other authors have highlighted that rapidly urbanizing cities, for instance in China cite{Cervero2008}, are "mimicking [suburbanisation] trends and patterns of the post-World-War-II United States". As such, we assume no severe structural deviations in respect to the modelled relationships between explanatory variables and building area prediction for these countries. The low feature importance for the variable "World Bank Region Code" seems to be in line with that assumption.

Reviewer comment	20. Table 5: How to calculate 'MSE' and 'MAE', and what are the units of these two measures?
Author	We've added further information in the Methodology section about how MSE and

Response	MAE were calculated. Methodology Section line 327: On the grid level, MAE describes the average of the absolute differences between predicted building area and reference building area obtained from authoritative sources or Microsoft's Global ML Building footprints in square kilometers. Accordingly, MSE describes the average of the squared differences between predicted building area and reference building area per grid cell. On the urban centers level, MAE describes the average of the absolute differences between predicted building completeness and reference building completeness. Accordingly, MSE describes the average of the squared differences between predicted building completeness and reference building completeness per urban center. These measures were calculated based on scikit-learn's python implementation\cite{scikitLearn2011}.
----------	--

Reviewer comment	21. Figure 7. Although the scatter plot shows a strong correlation, I notice that the distribution of residuals may be different for different completeness values. For instance, the higher the completeness value, the larger the residual. That means the completeness with a high estimated value may have a larger bias. It is needed to carefully explain the results.
------------------	--

Author Response	Thanks for this very important comment. We've added an explanation in the Methodology section. Methodology section line 343: For some regions, e.g. East Asia & Pacific or Sub-Saharan Africa, we observed heteroscedasticity in the distribution of residuals for urban centers. This means that the higher the predicted completeness value, the larger the residual and thus the uncertainty of our model. Still, our predictions are conservative in the sense that they are rather underestimating completeness for urban centers.
-----------------	--

Reviewer comment	22. Looking for some cities (e.g., Shanghai, Yogyakarta, Guilin) in the published dataset. The completeness of these cities are all equal to zero. But, there are a larger number of OSM building data, while checking in the OSM platform.
------------------	---

Author Response	Thanks for this very valuable comment and for making use of the data we provided. This is a very apparent lesson, why it is so important to provide access to research data and code. Based upon your comment we double checked our dataset and found that OSM building area statistics (and accordingly OSM completeness information) were accidentally missing for the following seven urban centers: Ajdabiya, Baalbek, Kharagpur, Mengzi, Zhuhai, Yogyakarta and Shanghai. We've fixed this issue and updated the data for the missing cities and the code in our GitHub repository.
-----------------	---

Reviewer 3

Reviewer comment	This article reports on spatio-temporal properties of crowdsourced geospatial data. Specifically, the core focus is on bias in OpenStreetMap. This is an interesting and worthy topic. The main conclusion drawn from the work is that users of OpenStreetMap need to be aware of the biases, notably cautioning against naïve use of the map. It is also good to see that the data sets and code used are being made available which should help ensure replicability etc. The work is set in the context of the UN SDGs but this aspect feels a bit of a tag on.
Author Response	Thanks for reviewing the manuscript and raising several aspects on how our study should be improved. First, we didn't want to convey the impression that the mentioning of the SDG's is just a "tag" that we've put on our work. Hence, we've put additional effort into making clear how urban OSM building data can support SDG monitoring. In the introduction section, we've created a new table (Table 1), which highlights policy and practice examples and applications of global urban analysis using building footprint data. This table contains examples that relate to SGD 11 (Make cities and human settlements inclusive, safe, resilient and sustainable) and SDG 3 (Ensure healthy lives and promote well-being for all at all ages) and SDG13 (Take urgent action to combat climate change and its impacts). In addition, we describe in detail in the introduction how building footprint data (from OSM) can assist monitoring of SDG 11. Introduction line 13: Building data is an essential asset in global urban analyses for assessing progress towards a number of important urban goals. For instance, SDG Indicator 11.3.1 ("Ratio of land consumption rate to population growth rate") would directly benefit from building footprint data. However this indicator is currently mainly based on easily available remote sensing data, e.g. World Settlement Footprint (WSF)\cite{Esch2022}, which exacerbates the monitoring of structural changes such as changes in floorspace per capita or re-densification trends. The monitoring of SDG Indicator 11.1.1 ("Proportion of urban population living in slums, informal settlements or inadequate housing") would benefit from an analysis of building blocks and street networks considering their spatial relations, such as density and neighbourhood relations\cite{Brelford2018}. SDG Indicator 11.7.1 ("Average share of the built-up area of cities that is open space for public use for all, by sex, age and persons with disabilities") can be partly derived from Sentinel-2 satellite data, but this data source falls short for distinguishing public from private green spaces as it fails to capture fine-grained urban structures due to its 10 metres resolution\cite{Ludwig2021}. Global Building Morphology Indicators could allow us to quantify the form of urban areas and enable novel analyses comparing morphological parameters across cities which go much further than existing approaches which are often based on aggregated population data\cite{Biljecki2022}. In addition to these examples of urban research using global building footprint data, Table \ref{table:use_cases} provides recent policy examples and practical applications of global urban analysis using building footprint data.
Reviewer comment	There are some concerns with the article. In particular, it does not really cover the content suggested by its title – it does not really explore or test in detail the digital divides that exist but rather reports on some well-known properties of

	OpenStreetMap. The scope is limited to completeness of building footprints. This has interest but there are many other dimensions that could be considered (e.g. provenance and data quality in terms of trust, credibility and accuracy etc.). The assessments of completeness are also limited. As just one concern, the size of the omission errors indicated by the recall statistic in Table 3 suggests that the data are not really suitable as a reference data set. In addition, the geographic bias in the data (noted on line 251) fits uncomfortably with the article's core aim and conclusion.
Author Response	This is indeed a very important comment. We've carefully adjusted the narrative of the manuscript and removed the notion of the term "digital divide", but now focus more on "completeness and inequalities in OSM". In the discussion section we removed the notion of "digital divide" as well. The title of the manuscript has been changed accordingly into "Investigating completeness and inequalities in OpenStreetMap: spatio-temporal analysis of global urban building data" to avoid the impression that our study covers the topic of the digital divide. We acknowledge that data quality is comprised of much more than completeness, however also highlight why we believe that completeness is essential to all examples and applications of global urban analysis in the introduction section. Introduction line 43: A common data quality requirement for many research and policy applications based on building footprint data is the completeness of the building stock for analysis purposes. This is particularly important for comparative analyses that seek to discern global urban patterns, such as for instance to derive a "global" dataset of critical infrastructure\cite{Nirandjan2022}, or to use "big data" for comparing urban morphology across the globe\cite{Boeing2021}. When unaccounted for, spatial bias in completeness can lead urban analysts and researchers to draw general conclusions which are only valid for well-represented (well-mapped) areas\cite{Meyer2022}. Completeness of building stock data is particularly important for ensuring equitable and fair decision making based on OSM data for the policy and practice applications of Table \ref{table:use_cases} and, as such, has a direct importance for the overarching principle of the SDGs of "leaving no one behind". We've checked to what extent external reference datasets could be used in our approach and utilized it to perform an independent assessment of the Microsoft Global ML Building Footprints dataset. Previously we could only report on the self-declared performance estimates by Microsoft. We have added Table 8 to the Supplementary Materials section of the manuscript, which reports precision and recall for the Microsoft building footprints by comparing it to the Geo-Wiki built-up surface validation dataset (10.1038/s41597-021-01105-4). We believe that recall values of 81%-86% (considering the World Bank regions) are sufficient for our modeling approach. Furthermore, this additional validation didn't unveil severe structural differences among World Bank regions in regard to recall. Results Section line 203: The quality of building footprint data from Microsoft (c.f. Supplementary Material Table \ref{table:MSquality_geowiki_validation}) showed a sufficient recall of more than 80% for urban centers in all regions. However, self-declared recall values by Microsoft have been considerably lower for some regions. This indicates that recall might be lower in rural areas, which have not been included into our study.

	Nevertheless, the remaining biases of the Microsoft buildings dataset for urban areas are also reflected in the results reported here and might lead to too low building area predictions and consequently too high OSM building completeness estimates. Methodology line 268: The Geo-Wiki built-up reference dataset cite{See2022} has been utilized to assess the suitability of the Microsoft building footprint dataset for our OSM completeness modelling approach. The Geo-Wiki campaign visually assessed very high-resolution satellite images of 50 K sample locations for the presence of built-up surfaces containing any building with a roof using a crowdsourcing approach cite{See2022}. All Geo-Wiki grid cells intersecting the urban center geometries for which Microsoft building data was available at the city level were considered in the analysis. Precision and recall for the Microsoft building footprints have been derived on the 10x10 meters Geo-Wiki grid level (see Supplementary Material Section Table \ref{table:MSquality_geowiki_validation}). Grid cells were defined as true positives (TP) if at least one Microsoft building footprint intersected with a Geo-Wiki grid cell labelled as 'Built-up'. True negatives (TN) were defined as Geo-Wiki grid cells labelled as 'Not built-up' that did not intersect with any Microsoft building footprint. Accordingly we defined false positives (FP) as grid cell labelled as 'Not built-up' which intersected with at least one Microsoft building and false negatives (FN) as the reverse. Finally, we also provide precision and recall for the Microsoft building footprints as self-declared by Microsoft in the Global ML Building Footprints GitHub repository (see Supplementary Material Section Table \ref{tab:MSquality}).
--	---

Reviewer comment	The basic issues connected to the spatial and temporal biases in OpenStreetMap are also well known. This is particularly the case with the relatively easy to assess issue of building completeness. A quick search for papers on building completeness in OSM shows >4,000 articles in the last 4 years. The nature of OpenStreetMap contributors is also well known (e.g. they tend to be male, well-educated etc.) and the patterns of contribution (e.g. often on 'pet' topics, near home etc.) have been widely reported. Geographic biases, from basic urban v rural contrasts to regional scale imbalances are all very well known. Many studies have shown that the data for Europe and North America are more complete than elsewhere. Some other locations also have high completeness, especially associated with disasters (e.g. the OSM data for Haiti after the 2010 earthquake is a classic example used to promote OSM). The recent trend to bulk updates, often from people/organisations that can be regarded as relative expert contributors, is also well known.
------------------	---

Author Response	Thanks for providing this critical perspective on our work. We've tried to make clear how our work is different from existing studies in the manuscript. Although there have been studies in the past which investigated OSM building completeness, we do not share the opinion that the temporal and spatial biases in OSM are investigated sufficiently at the global scale nor that the spatial footprint of these biases can be considered well known. Our study is among the first, which not only reports on the general biases in OSM (such as global north - global south), but makes these biases visible at a very fine spatial scale of 1 x 1 kilometer for about 13,000 urban centers. We believe that it is the particular strength of our study to clearly depict what the "elsewhere" you mentioned is and where these urban centers are located and how completeness is distributed within these cities. We have invested additional efforts to make these novel completeness maps
-----------------	---

	available via the ohsomeHex platform (https://hex.ohsome.org/#/urban_building_completeness), which allows users to interactively explore the completeness at the urban centers level and the grid level for the entire time range covered by this analysis (2008-2023). All data is furthermore available in GitHub (https://github.com/GIScience/global-urban-building-completeness-analysis) and on Figshare (https://figshare.com/articles/dataset/Dataset_Investigating_completeness_and_inequalities_in_OpenStreetMap_-_spatio-temporal_analysis_of_global_urban_building_data/22217038) To the best of our knowledge, as of 2023-01 there exists only a single study (which has been published in 2022-12 after we've submitted our manuscript) which investigates the topic of urban OSM building completeness at a global scale (10.1080/17538947.2022.2159550). Indeed, there are other studies that investigate the completeness of the OSM road network and land use land cover information in OSM at a global scale. We've set our results into context with these studies in the discussion section. Discussion section line 166: Our results confirm the similar, albeit less dichotomous, global completeness pattern of land use land cover (LULC) information from OSM\cite{Zhou2022} and stand in slight contrast to the relatively higher and more evenly distributed completeness for OSM's road network\cite{BarringtonLeigh2017}. This highlights the complexity of OSM mapping activities and the challenge for urban analyses based on OSM data. Discussion section line 219: Our findings extend the general pattern of urban OSM building completeness as of 2020-01\cite{Zhou2022b} a) by highlighting the temporal evolution of mapping activity in conjunction with other events such as the COVID-19 pandemic, b) by considering a spatially much more balanced and extensive training data set from various sources and c) by investigating the global pattern and consequences of inequalities in completeness. In addition, geospatial data quality is comprised of dimensions beyond measuring completeness\cite{Oort2006}. For some sectors, such as public health programs, assessment of completeness is only the first step, and information on building usage is also required, but often only available for a small subset \cite{Sturrock2018}. It has been shown for LULC in OSM that the spatial pattern for completeness and accuracy are not necessarily the same\cite{Zhou2022}. Whereas urban OSM building completeness and OSM LULC completeness show similar global trends on the national level, OSM building data accuracy needs to be further investigated, e.g. towards its potential utilization as training and/or validation samples in machine learning models.
--	--

Reviewer comment	The danger of using biased data is well known. The danger of using an average value (line 58) is really rather obvious and nothing more than common sense. Also although an aim is to determine if the data are 'good enough' (line 40) the authors never state what level of quality is required. The latter is also highly application dependent so for some users the data will be perfectly useful while for others they would be inappropriate.
Author Response	We agree that most researchers are aware of the challenges that come with using biased datasets. We've removed the apparently superfluous notion in the results section that "these regional differences illustrate that the global average is of limited explanatory power."

Furthermore, we believe that our study results will be very valuable for these researchers using OSM building data, as our precise quantification of the uneven coverage in OSM will help to design their studies and models thoroughly. We highlight this aspect in the discussion section.

Discussion section line 172:

First, the unequal patterns of completeness we evidenced make it essential for urban analysts to assess the potential negative impact of missing data, i.e., OSM data users should investigate if the intended urban analysis is subject to spatial bias caused by OSM's uneven spatial coverage at multiple scales.

To support this assessment, we provide an open dataset\cite{Herfort2023} resulting from this study with estimated completeness maps for 13,189 urban agglomerations worldwide using a 1x1 kilometer grid, which enables the assessment of variations within and across urban centres.

We encourage researchers and practitioners to further develop this datasets and the other data quality assessment methods which may help to quantify the highly uneven geographies of participation in communities such as OSM or as observed in Wikipedia\cite{Graham2015}.

Global studies and global frameworks will benefit from these approaches, as researchers will be able to draw more robust conclusions and avoid misleading recommendations for decision makers once the biases in OSM's coverage are known and can be accounted for.

It is correct that data quality and what can be considered “good enough” is highly application dependent. Data needs to be fit-for-purpose. As such, we admit that we can't state what level of quality is required for global urban analysis applications in general. Thus, we've carefully rephrased our research questions to better account for this. We believe that the users of the OSM data, which know the purpose for which it should be used, are in the position to decide if the dataset meets their requirements using the results from this study and the additional datasets and materials that we provide.

Introduction line 54:

RQ1: What is the completeness of OpenStreetMap building data in the context of global urban analysis applications?

RQ2: How unequal is urban OpenStreetMap building data distributed within the space of a city, across continents and on the global level?

REVIEWERS' COMMENTS

Reviewer #1 (Remarks to the Author):

The authors have considered all my comments carefully, and improved the paper accordingly. Thank you. In particular, I appreciate the modification of the title, the research question, and the addition of Tables 1 and 3. Further, it appears that the authors have made considerable effort to refresh the analysis with a recent OSM snapshot (incl. a recent one by Microsoft's dataset) to keep the results as new as possible, against the backdrop of OSM's continuously evolving content and completeness.

Admittedly, I cannot check the revision in detail since I was not provided with a track changes file, but reading the response letter and checking the new version of the paper, I think that the paper reached a commendable level of quality and the only remaining comments I have are minor.

1. While this paper remains a major advancement in this topic, and a high quality article which deserves to be published, there have been some new related papers in assessing building data quality of OpenStreetMap at a very large scale or use building data from OSM at a massive scale with some discussions on quality:

<https://doi.org/10.3390/ijgi12040143>

<https://doi.org/10.1038/s41597-023-02040-2>

<https://doi.org/10.1016/j.buildenv.2023.110295>

The authors may want to regard these new papers to have their review as up-to-date as possible like their analysis.

2. Figure 10 is a great addition to the paper, but the choice of the colour scheme is questionable. This map shows data on a diverging scale (with 1 in the center), but the colour palette used to represent it is not a diverging one (in fact, it does not appear to be one to represent numerical data, rather categorical).

3. Thanks for clarifying that the grid was adopted from GHS-UCBD. I understand that this is a '1x1 kilometer grid based on the equal-area Mollweide projection'. Doesn't that mean that the size is not always 1x1 km? Looking at the website of the project, at higher latitudes, the cells are not square. Has this been taken into account? Maybe a discussion point if it affects the method?

Reviewer #2 (Remarks to the Author):

The authors have replied to all my comments and revised the manuscript accordingly. I suggest to publish the paper.

Response to Reviewer Comments

Reviewer 1

Reviewer comment	1. While this paper remains a major advancement in this topic, and a high quality article which deserves to be published, there have been some new related papers in assessing building data quality of OpenStreetMap at a very large scale or use building data from OSM at a massive scale with some discussions on quality: https://doi.org/10.3390/ijgi12040143 https://doi.org/10.1038/s41597-023-02040-2 https://doi.org/10.1016/j.buildenv.2023.110295 The authors may want to regard these new papers to have their review as up-to-date as possible like their analysis.
Author Response	We considered new publications in the discussion section of the manuscript. “Recent work has outlined pathways towards regional and global scale analysis of the quality of building attribute data from OSM and other sources cite{biljecki_quality_2023, milojevic-dupont_eubucco_2023}.”

Reviewer comment	2. Figure 10 is a great addition to the paper, but the choice of the colour scheme is questionable. This map shows data on a diverging scale (with 1 in the center), but the colour palette used to represent it is not a diverging one (in fact, it does not appear to be one to represent numerical data, rather categorical).
Author Response	We have adjusted the color schema in Figure 10 and used a diverging scale. Please note that this figure is now provided as Supplementary Information (Supplementary Figure 2).

Reviewer comment	3. Thanks for clarifying that the grid was adopted from GHS-UCBD. I understand that this is a '1x1 kilometer grid based on the equal-area Mollweide projection'. Doesn't that mean that the size is not always 1x1 km? Looking at the website of the project, at higher latitudes, the cells are not square. Has this been taken into account? Maybe a discussion point if it affects the method?
Author Response	We have adjusted the description of the grid size and shape. “Each urban center was spatially disaggregated using a one square kilometer grid based on the equal-area Mollweide projection. The grid adopted the same structure utilized by the raster datasets of the GHS-UCBD. The grid cells are not always squared, as they deviate from a perfect rectangular 1x1 km shape depending on latitude and longitude of each grid cell. The shape distortion adds uncertainty to our results for a very small number of urban centers. These are located in very low and very high latitudes which are also far away from the Greenwich meridian, e.g. in New Zealand.”

Reviewer 2

Reviewer 2 did not provide further comments for improvement.